

# FLOOD RISK RELATED TO A FLUVIAL SYSTEM MODIFIED BY DAMS WITH EMPHASIS ON MORPHODYNAMIC AND HYDROLOGICAL ASPECTS

Karina Vanesa Echevarria[1,2], Susana Beatriz Degiovanni[2], Mónica Teresa Blarasin[2], Carlos Eric[2], María Jimena Andreazzini[1,2]

[1]The National Scientific and Technical Research Council (CONICET), Argentina

[2]National University of Río Cuarto. Ruta Nacional N° 36 km 601, X5804BYA, Río Cuarto, Córdoba, Argentina.

*Correspondence to* Karina Echevarria (karyechevarria@yahoo.com.ar)

## Sect 1. Abstract

Villa Dolores and peripheral localities (Córdoba province, Argentina) flood risk was analyzed. They have expanded their urban area on the inactive channel of De Los Sauces River after the dam construction. Downstream receives water from 5 sub-basins whose upper basins, located in Mountains, generate the largest flash floods affecting these locations. The risk was analyzed considering the hazard in three different threat scenarios and vulnerability. The areas of greatest risk are reduced and they are confined to urbanized fluvial belt sectors in historical floodplain and low terraces. Geomorphological studies were effective for risk estimation, being irreplaceable in the hazard mapping.

## Sect 2. Introduction

The occurrence of floods is the most frequent natural disaster, affecting both rural and urban settlements. Flooding is a global phenomenon which causes widespread devastation, economic damages and loss of human lives (Jha et al., 2012).

For many years, the dam construction has been a common practice in the fluvial systems management, for water supply, irrigation, hydroelectric generation and flood control. These practices produce important hydrological, sedimentological and morphodynamic changes in the modified watercourses (Vericat and Batalla, 2004; Graf, 2006, Gregory, 2006, Schmidt and Wilcock, 2008, Ma et al., 2012, Grant, 2012, Xia et al., 2016), generally decreasing their discharge, sediment load and erosion and flood hazard towards downstream. Assuming these conditions as infallible, induces the advance of urbanization on the alluvial plains, which leads to a potential increase of flood risk during extraordinary events, mal function or dam failure operation, among others (Dewan et al., 2007, Bosisio, 2011). The flood risk associated to fluvial systems has increased in the last decades in most countries worldwide due to the absence or inefficiency of the land use plans, especially those related to the urban expansion in the alluvial plains (Tucci Morelli, 2007, Vidal and Romero, 2010, Jha et al., 2012, Sayed and Haruyama, 2016). In Argentine, and linked to an increase in rainfall, floods caused by rivers controlled by dams have been recurrent in the last years, which produced important damages in the cities located in the river margins (Graneros,



La Madrid y Alberdi -Tucumán, 2015; Río Tercero, Villa María, Bell Ville - Córdoba, 2014, 2015; Limay river- Río Negro).
Especially in the Córdoba province the major rivers whose upper basins are developed in the Pampeans Mountains (Suquía,
Xanaes, Ctalamochita, Cruz del Eje, De Los Sauces, among others) have been intervened with dams since the first decades
of the last century to generate electricity and, secondly, to store water for irrigation, drinking and recreational purposes.
Although there are some studies related to the problems associated with flooding from these rivers (Barbeito and Ambrosino,
2004, Barbeito et al., 2004, Echevarria et al., 2017), there are still few cartographic works on flood hazard and risk.
Regarding the methodologies used to predict flood risk, the studies based on the mathematical treatment of idealized
parameters are the most used (Baker, 1994). They assume that they have similar real events, whose results are imposed on
society through engineering designs, zoning of flood hazards, among others. In contrast to this perspective, the
geomorphological studies make a more comprehensively approach to floods (Sayed and Haruyama, 2016), considering
sedimentological, morphological, and paleohydrological records, among others, and larger spatial and temporal scales
(Garzón Heydt, 1985; Schumm, 1977, 2005; Baker, 1986, 1994; Baker et al., 1988; Benito y Thorndycraft, 2005). The
complementation of conventional hydrological approaches and structural and non-structural measures (Dewan et al., 2007,
Masood and Takeuchi, 2012) arising from geological-geomorphological studies, allow the development of more
comprehensive hazard and risk flood maps, which would favor an adequate territorial planning.
In relation to environmental maps different meanings of the concept of risk are used (Hermelin, 1991; Panizza, 1992; Bosque
Sendra et al., 2005; Kron, 2005 Fedeski and Gwilliam, 2007; Merz, et al., 2007; Vilches and Reyes; 2011, Field et al., 2012,
Koks et al., 2015, among others) and therefore they may have different readings. In this work we use the definition of
Panizza (1992) who considers risk as the product of the interaction of the hazard derived from a natural process with the
vulnerability of the anthropogenic environment. Using this conceptual basis, the flood risk of the city of Villa Dolores and
peripheral localities (Córdoba province, Argentina – Fig. 1) was analyzed. This objective was planned taking into account
that the mentioned towns have expanded their urban areas and services on the inactive channel of the De Los Sauces river
after the Medina Allende dam construction in 1942.



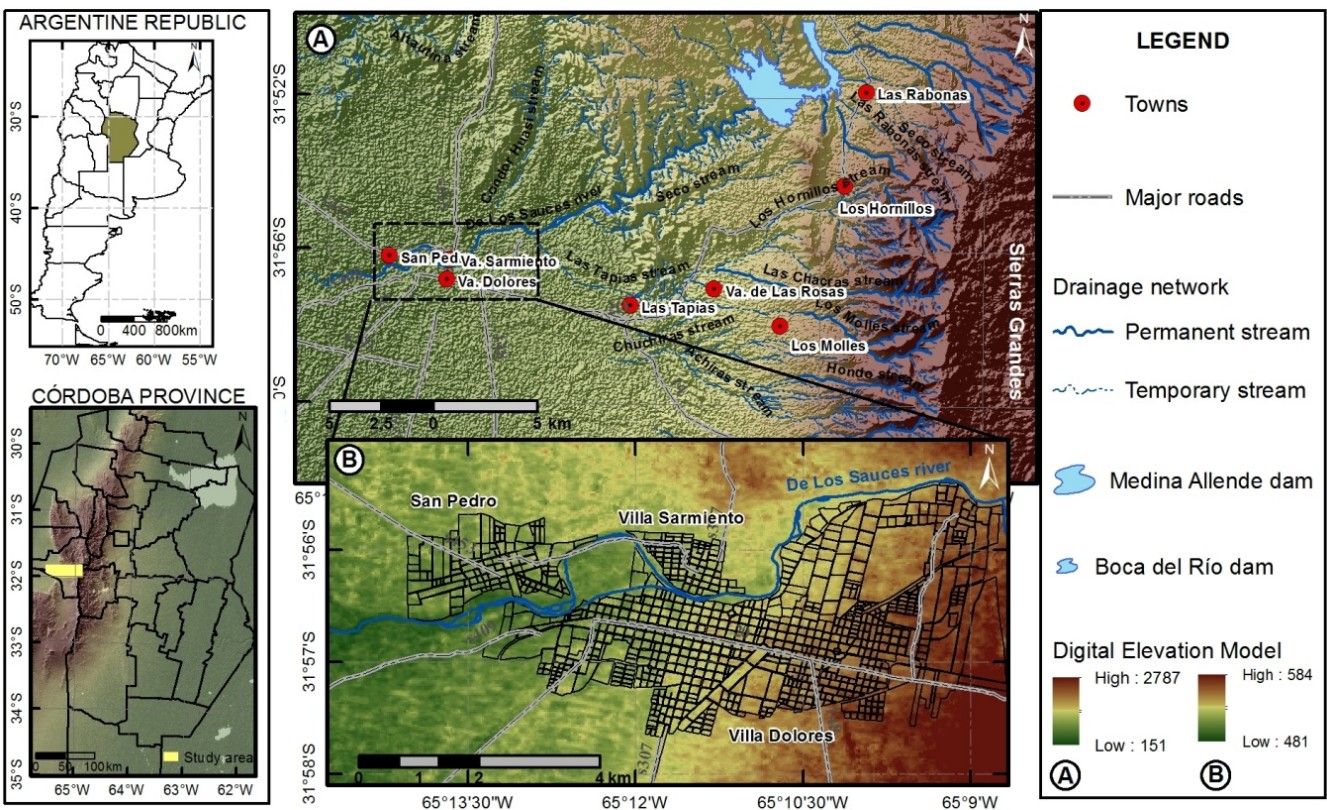

**Figure 1.** Location of the study area. Digital elevation model from SRTM (Shuttle Radar Topography Mission - data available from the U.S. Geological Survey).

## Sect 3. Regional setting

The study area is part of the Pampean Mountains geological province of Córdoba (Ramos, 1999) which exhibit a classic tectonic setting defined by regional faults that control the outstanding elements of the landforms. Towards the East the Grandes and Pocho Mountains stand out. These are composed by metamorphic and granitic rocks (Precambrian-Lower Paleozoic), where the highest altitudes (maximum height 2,900 masl) and slopes (maximum 30 %) were observed. Next to the mountainous front, two levels of alluvial fans of Pleistocene and Holocene age (Bonalumi et al., 1999) are developed, forming strongly ondulating reliefs (maximum slopes on the order of 10 %), which show neotectonic activity evidences. This piedmont environment is formed by conglomerates with clasts of very variable sizes, with a psefitic to sand-silt matrix, covered in the middle distal sectors by loess and/or fluvio-aeolian materials (Fig. 2). A noticeable drainage network has been developed in these alluvial fans, nowadays with very deepened courses.

The western zone of the study area is part of a major intermontane depression (Quines-Ulapes-Chancani) and is constituted, mainly by Cenozoic fluvial sequences of varied energy pertaining to the alluvial fan of De Los Sauces River and, secondarily, by Quaternary aeolian loessic and sandy sediments (Fig. 2).



The climate of the zone is semiarid mesothermal (Thornthwaite, 1948). Average annual precipitation is 628.2 mm yr$^{-1}$ for
1961-2014 period (Villa Dolores weather station– National Meteorological Service). However, there are variations in the
precipitation values due to landform. In this way, the rainfall gradually decrease from the mountains towards the plain
(Gorgas et al., 2003). Approximately 77 % of the precipitation is concentrated in the spring-summer period and is
responsible for the largest flood events associated with the watercourses.

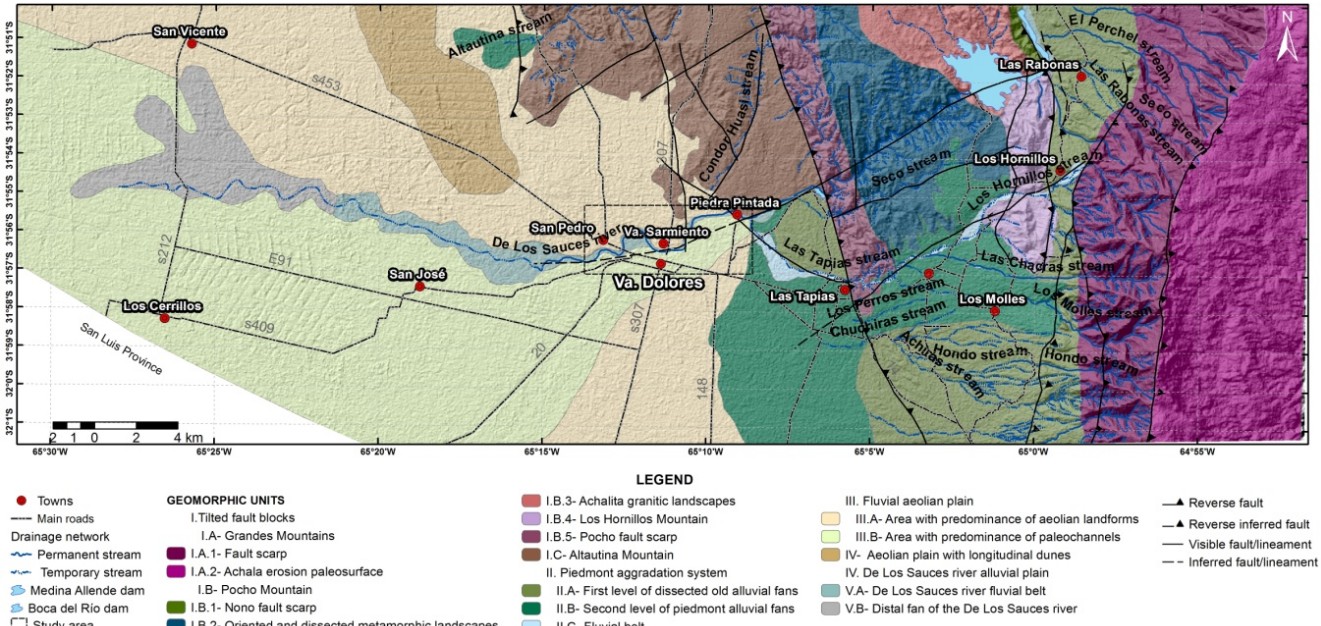


**Figure 2.** Geological-geomorphological map.

The De Los Sauces River drains an area of approximately 1,200 km$^2$. The upper and middle basin are developed in the
Grandes and Pocho Mountains and the lower basin in the western intermountain depression. This course starts in the
confluence of the Panaholma and Mina Clavero rivers and its middle reach is intervened by the Medina Allende and the
Boca del Río dams. Downstream of the dams the river receives water from five important sub basins whose upper basins,
located in the Grandes and Pocho Mountains, generate the largest flash floods that can upset the studied area (Fig. 3).





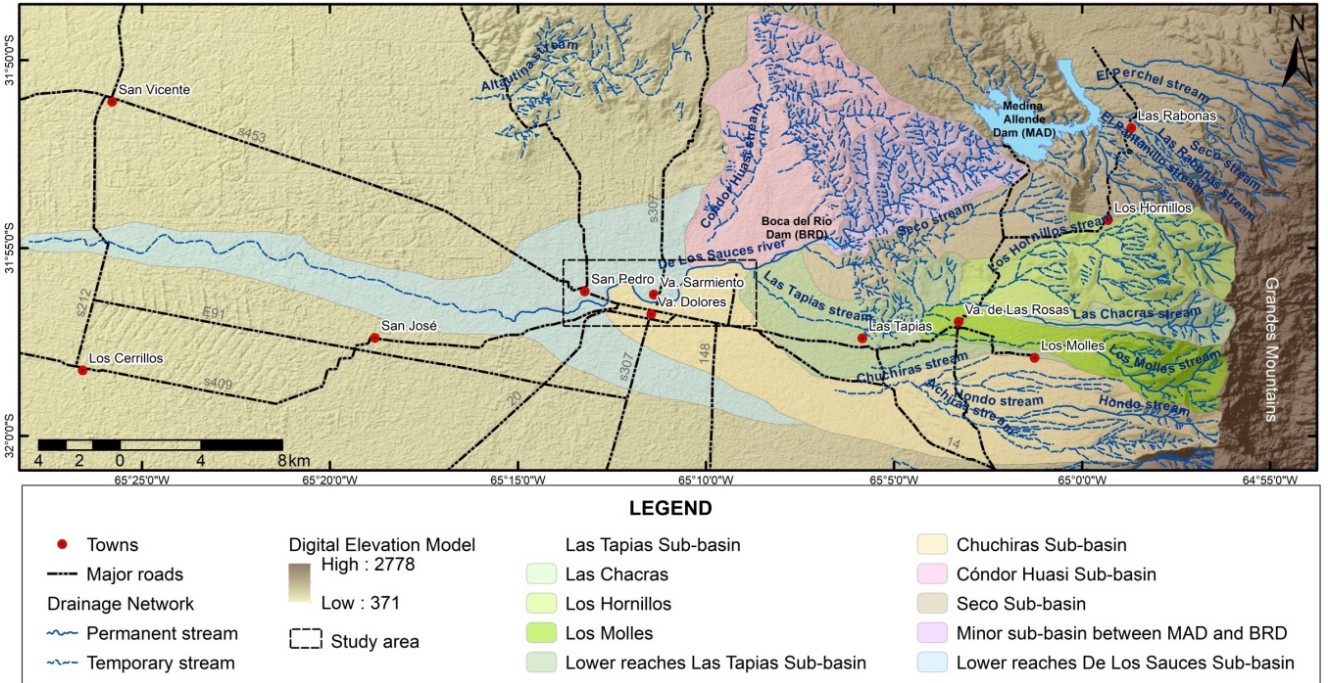

**Figure 3.** De Los Sauces basin map.

**Sect. 4. Methodology**
In this work the risk concept of Panizza (1992) was used which is synthesized in the following equation:
$$RISK = HAZARD \times VULNERABILITY$$
The **Hazard** represents the *Susceptibility* or natural fragility of a region exposed to a certain Threat. The susceptibility
includes the geological, geomorphological, lithological, hydrological, geotechnical aspects, among others, that together
determine the behavior of an area in front of a natural process (Panizza, 1992), whereas the *Threat*, according to Hermelin
(1991), is the probability of occurrence of a potentially destructive phenomenon within a specific time period for a specific
area. Finally, the **Vulnerability** includes the population aspects, social organization, economy, programming, cultural,
historical and natural values of interest for the preservation (Panizza, 1992, Cendrero, 1987).
Currently there is a wide range of procedures adopted for the realization of flood maps with the use of GIS tools (Domínguez
Chávez et al., 2015).
In this work, layers overlay tools in vector format for the elaboration of cartography at a detailed scale were used.
The Villa Dolores topographic maps, scale 1: 50,000 (National Geographic Institute), aerial photographs from1970
(approximate scale 1: 20,000), satellite images of the Google Earth software were used. However, and taking into
consideration the scarce accuracy that the satellite images have for this detailed work, the field survey data was the main
input for this study. Thus, water-level marks (considering vegetation, sediment distribution, erosion features and witness




reports), overflow sites, morphometric data of cross fluvial sections, population features and land uses were surveyed and
population surveys were made.
For the analysis of the flood **susceptibility**, the geomorphological characteristics of the area were taken into account,
considering the topographic aspects (elevation, slope). Particularly, the morphological and morphometric changes of the
main river channel as a result of the Medina Allende dam operation were evaluated. The previous scenario was reconstructed
from aerial photographs from 1970 and the field survey evidences.
To assess the **threat**, three flood scenarios of different magnitude and recurrence were considered, resulting from the
combination of discharges controlled by the partial or total sluice gates opening in the Medina Allende dam and the flow
coming from the not intervened sub basins. For the flood transport assessment, land use types in the channel and in the flood
plain were taken into account, especially the percentage vegetation cover and the activities and structures that affect the flow
distribution and the roughness coefficient.
Due to the lack of records in the courses without interventions the flood flows were estimated for the extraordinary event on
February 4, 2014 using the Manning equation (Chow, 1994):

$$v = \frac{R^{\frac{2}{3}} * S^{\frac{1}{2}}}{n}$$

Where: R is the hydraulic radius (A/Wp), A is the cross section area, Wp is the wet perimeter, S is the channel slope
(calculated for a stream reach from contour lines from topographic maps) and n is the roughness coefficient. Based on
channel and floodplain characteristics (dominant particle sizes, type and percentage of cover vegetation, etc.) a "weighted n"
was established (Chow, 1959).
The flood flows derived from the opening of sluice gates were informed by official sources (Epec- Provincial Energy
Company of Córdoba).
To estimate the **Vulnerability**, the density of houses in urban and rural areas and the vial infrastructure (roads, bridges and
accesses) were considered.

**Sect. 5. Results**
**Subsect 5.1. Geomorphological and Topographic Characterization**
The study area is located in the proximal sector of the alluvial paleofan (Neogene-Quaternary) of the De Los Sauces River,
where the current course presents different incision degree and varied development of the fluvial belt.
In this context, five geomorphological units were recognized (Fig.4).



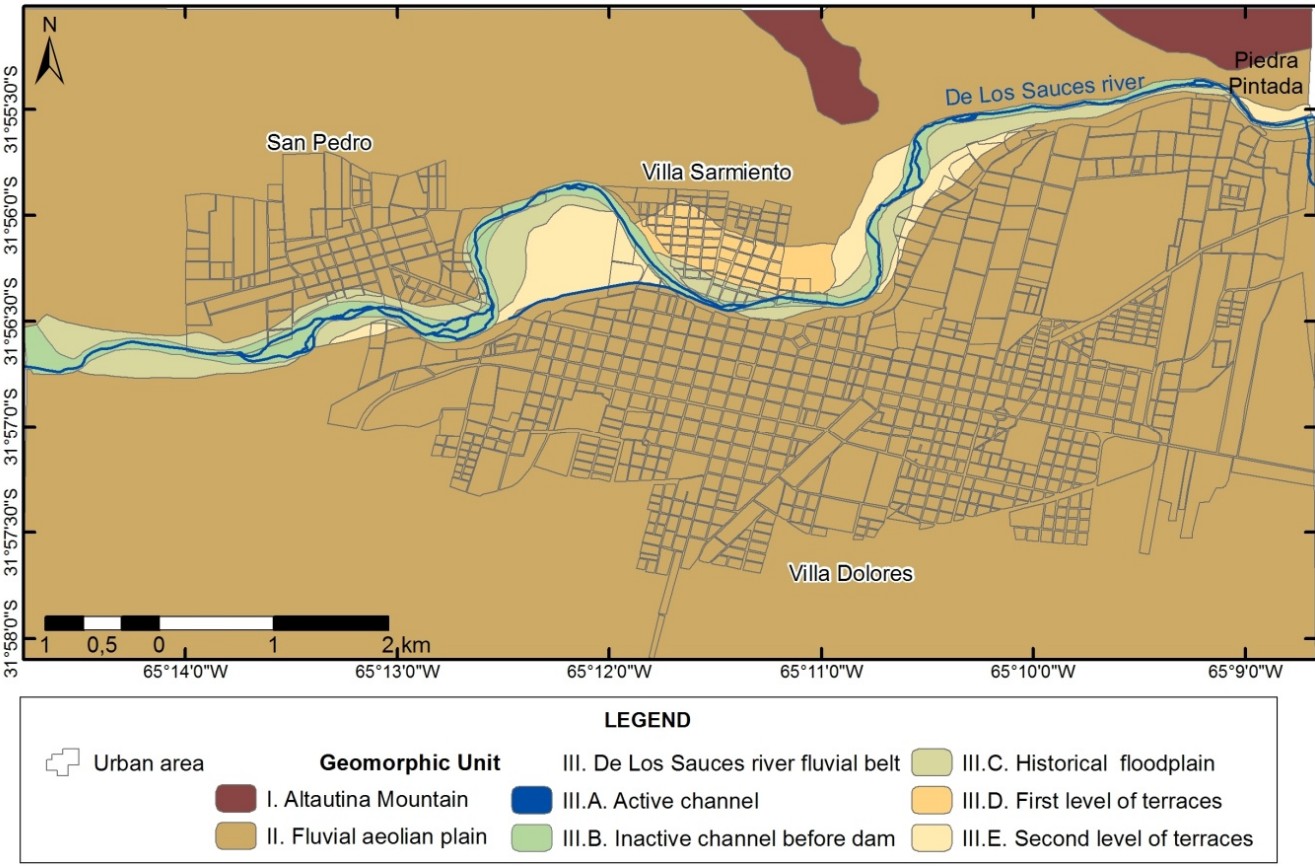


**Figure 4.** Geomorphological Map of the study area


**I-Fluvio-aeolian Plain:** corresponds to the oldest surface of the paleofan. The relief is very gently undulated, where
the loessical layers and longitudinal dunes are interdigitated and/or overlaying the paleochannels and overflow lobes of the
De Los Sauces River. It has a slope to the west on the order of (0.55) and a height with respect to the active channel that
decreases in that direction from 8 to 2-3 m approximately when entering the middle sector of the paleofan.
**II-De Los Sauces River fluvial belt:** it extends downstream from Piedra Pintada location and is the result of
different incision pulses from De Los Sauces River during the Upper Holocene to the Present. It has a width between 300
and 1,500 m, associated with straight (bedrock) and meandering (alluvial) channel reaches. It includes two discontinuous
levels of terraces (III.D and E-Fig. 4) and a small floodplain (III.C-Fig. 4) associated with the active channel (III.A-Fig. 4).
The oldest terrace level (T1) has a slope on the order of 3-4 m and the lower level (T2) of 2-3 m.
The channel of De Los Sauces River shows variability, not only linked to geological controls but as a result of the operation
of the Medina Allende dam. In general, the bedrock segment do not exhibit changes, while the alluvial channel lost its
braided behavior, although it maintained its sinuosity, prevailing a semiconfined single channel with and erosive behavior.





The channel width was reduced up to 85 %, generating a historical floodplain. The channel was segmented in three parts
considering the most relevant morphological and morphometric characteristics in pre and post dam conditions (Table 1).

**Table 1.** Most relevant morphological and morphometric characteristics of the channel in pre and post dam conditions.

| U.II.3 Active Channel | | Types of river channel | Channel Patterns | Height Bank (m) | Length (km) | Slope (%)† | Width (m) | Width channel reduction (%) 1970-2017 |
|---|---|---|---|---|---|---|---|---|
| R1 | Pre-dam | Bedrock-Alluvial | Straight Single Channel (SI:1.1) | 3-4 | 3.5 | 0.5 | 30 | 55 |
| | Post-dam | | | | | | 16 | |
| R2 | Pre-dam | Alluvial Gravelly-Sandy | Meandering with overlay braided, mobile bars | 3-6 | 5.8 | 0.32 | 40 | 80 |
| | Post-dam | Alluvial Medium/fine Sandy | Meandering (SI: 1.6) Single channel dominate and secondary channel, locally (BI: 2), very vegetated and stable bars | | | | 8 | |
| R3 | Pre-dam | Alluvial Gravelly-Sandy | Meandering Braided | 2-5 | 4 | 0.32 | 70-60 | **85** |
| | Post-dam | Alluvial Medium/fine Sandy | Multichannel (SI: 1.2, BI: 4) Irregular, erosive and secondary channels. Ponds presence | | | | 10-12 | |

SI: Sinuosity Index, BI: Braiding Index

**Subsect 5.2. Hydrology and Hydrometry**
In the hydrographic map the medium and low reach of the De Los Sauces River, downstream of the Medina Allende dam,
and also Las Tapias and Chuchiras streams are shown (Fig. 3). These courses drain the western scarp of the Grandes
Mountains and have a torrential regime controlled by lithology, high slopes and summer rainfall intensity. In the piedmont
sector, although the main collectors are incised in inactive alluvial fans, avulsion processes and transfers to neighboring
basins can be registered in extraordinary floods, attenuating then the flood peaks. These sub basins are not instrumented
therefore there is no systematic discharge records. Table 2 presents data corresponding to instantaneous gauging (2011 and



2014) and flood discharge estimates with a recurrence of approximately 25-30 years (according to journalistic information
and witnesses.

**Table 2.** Hydrological and hydrometric characteristics of the main sub-basins

| Subbasin | Area (km$^2$) | Discharge (m$^3$s$^{-1}$) | | | Regime |
|---|---|---|---|---|---|
| | | Dry Period | Wet Period | Extraordinary Flood | |
| **Las Tapias stream** | 116 | 0.01 | 0.02 | 129 | Temporary |
| **Los Hornillos stream** | 45 | 0.006 | 0.02 | 25.1 | Permanent/Temporary |
| **Las Chacras stream** | 11.4 | 0.003 | 1.08 | 24 | Permanent |
| **Los Molles stream** | 21.5 | 0.011 | 1.17 | 30.1 | Permanent/Temporary |
| **Chuchiras stream** | 80.5 | 0 | 0.29 | 200 | Temporary |
| **Hondo stream** | 12 | 0.023 | 1.41 | 13.2 | Permanent/Temporary |
| **Achiras stream** | 15 | 0.0104 | 0 | No data | Temporary |
| **De Los Sauces river** | 390 | 0.14 | 0.18 | 130 | Permanent/Temporary |


In the De Los Sauces River, the only equipped station (La Viña Dam), located immediately downstream of Medina Allende
dam, belongs to the National Water Resources Secretary and has been in operation from 1928 to 1980. The average
discharge for this period is 5.6 m$^3$ s$^{-1}$, with a maximum of 900 m$^3$ s$^{-1}$. The Medina Allende dam has a storage capacity of 230
hm$^3$ and, through 8 sluice gates, it can evacuate an extreme discharge of 1,200 m$^3$ s$^{-1}$ (EPEC, 2009). Lower discharge values,
on the order of 30-40 m$^3$ s$^{-1}$, were registered in 2015 and 2016 by the partial opening of 4 sluice gates.

**Subsect 5.3. Land Use:** the defined units are presented in Fig. 5 and the main features are described below:
**1) Urban Areas:** The unit includes the Villa Dolores, Villa Sarmiento and San Pedro towns located on both margins of the
De Los Sauces River. They show an important population growth (130-180 %) in the last 50 years (NU. CEPAL. CELADE,
2001). Although the expansion of these localities has been carried out in several directions, a moving towards the historic
floodplain can be observed (Fig. 4), as occurred in the Villa Dolores western sector where the "Paso de la Virgen" densely
populated neighborhood was placed. This leads to human impact increases such as waste disposal sites, soil infiltration
capacity reduction, naturalness and functionality decrease (especially by loss of vegetation cover), water contamination,
among others.
**2) Agricultural - Horticultural**: It extends mainly in the fluvio-aeolian plain (Fig. 4) covering most of the study area. This
rural area has a very low population density. In general, the crops occupy small extensions and are irrigated through canals
and/or ditches. Secondly, extensive livestock farming was observed.
**3) Recreational:** As it is observed in Fig. 5 all the recreational uses are located in the channel and the floodplain of the De
Los Sauces River. Due to high pressure from land use in the summer period and the moderate to low vegetation cover
degree, the Piedra Pintada bathing site highlights (Fig. 6). In this river reach, there are different facilities across the channel




(a bridge, a ford, a small dike and a duct). A second bathing site is located immediately upstream of the bridge that connects
Villa Dolores and Villa Sarmiento. This sector has scarce to null coverage of shrubs and trees, is heavily modified with
channels, a swimming pool and roads, exhibiting significant environmental degradation.

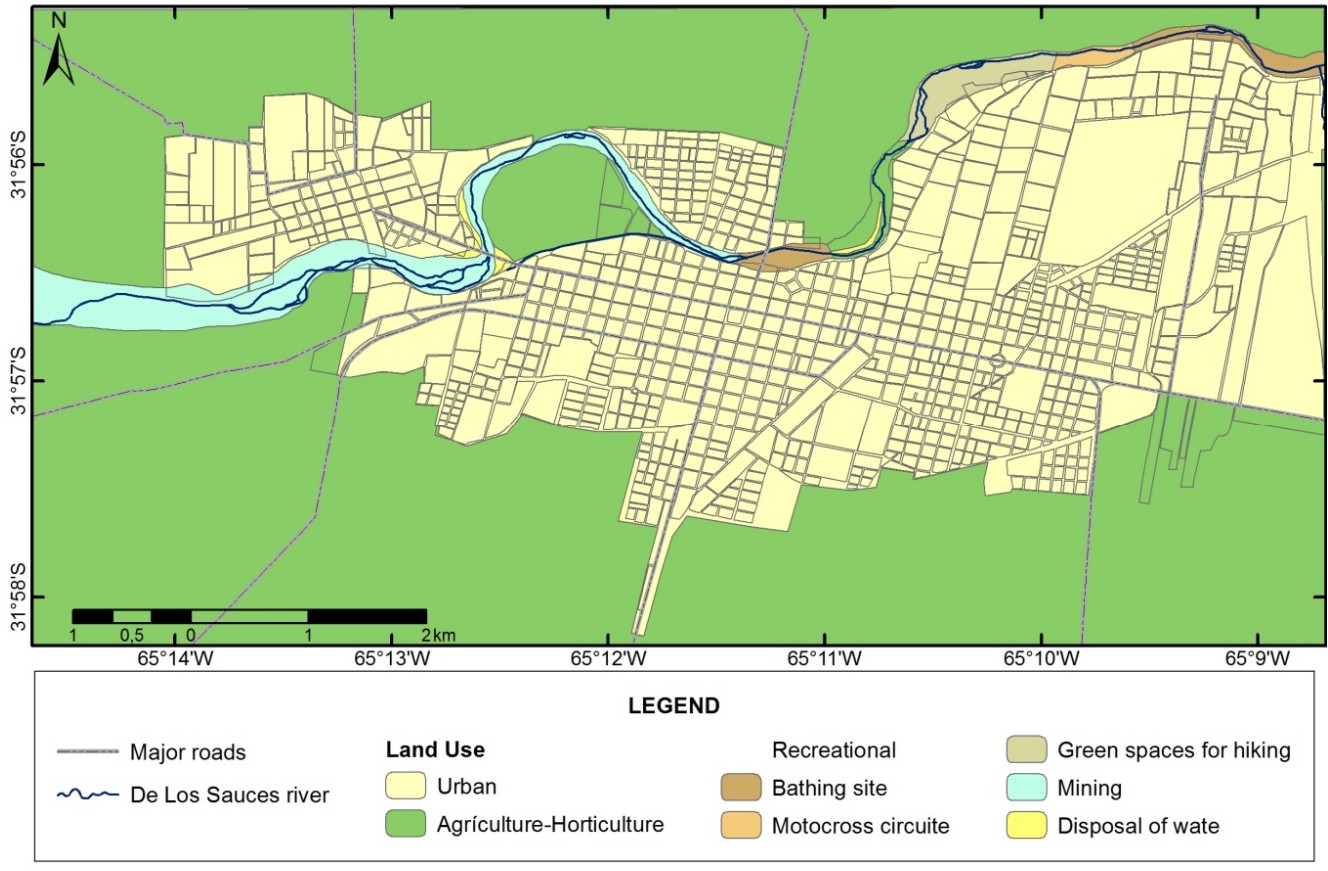

**Figure 5**. Land use map.

A *motocross circuit*, which generates local changes in the floodplain relief, interferes with the water flow distribution and
also green spaces planned for *hiking* complements the recreational activities. These are the areas that show scarce human
modifications and that preserves the highest percentage of tree species.
**4) Mining:** the handheld extraction of sediments in the river channel and the floodplain of the De Los Sauces River is the
main mining activity. It is carried out from the Villa Dolores towards downstream being the San Pedro area the more
intensive exploited (Fig. 6). This activity generates a change in the channel morphology, causing changes in fluvial dynamics
and a strong impact on the landscape quality.
**5) Disposal of waste:** the dumping of illegal waste along the fluvial belt is a common practice, although three sites of greater
relevance were detected. This activity is associated with the expansion of urbanized areas and the loss and null conservation
of natural spaces (Fig. 6).





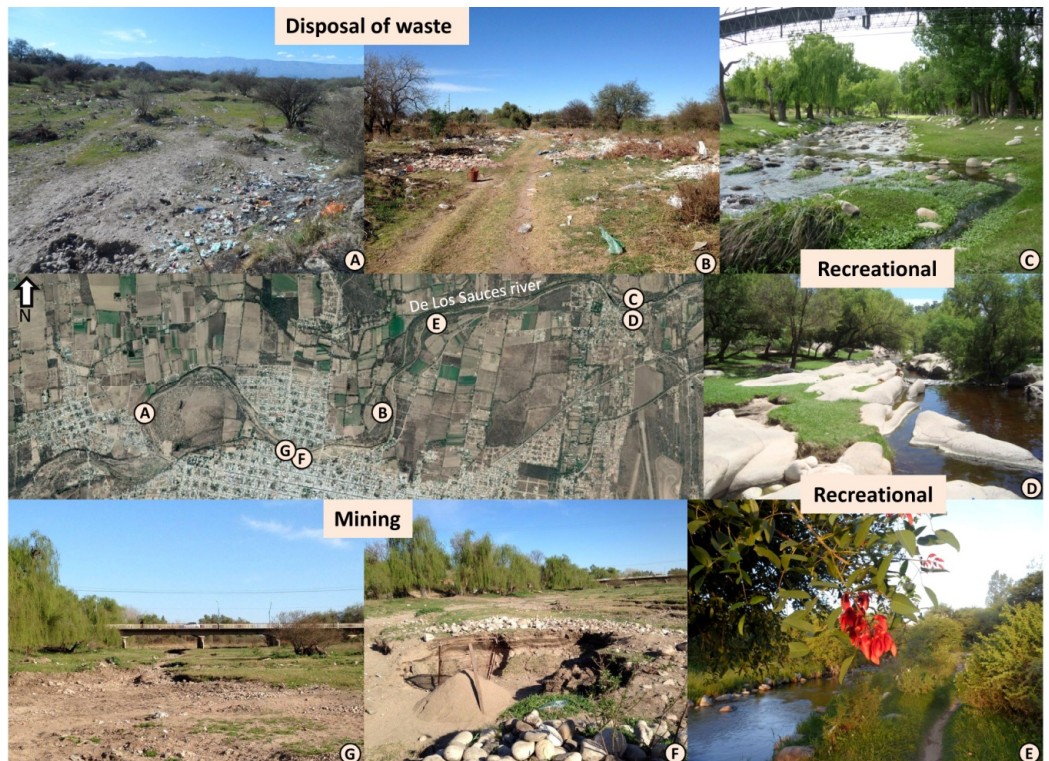

**Figure 6.** Photos of different land uses in the fluvial belt of the De Los Sauces River.

**Subsect 5.4. Analysis of flood risk**

**Subsect 5.4.1 Fluvial flood hazard**

-**Susceptibility analysis**

Five susceptibility classes were defined (Table 3) which were evaluated in each geomorphological unit (Fig. 7). As can be observed in the map, the susceptible zones are those located in the most modern fluvial belt. Taking into account that it is incised in the paleo alluvial fan and then deepened, these zones have very low susceptibility.

**Table 3.** Susceptibility classes evaluated for each geomorphological unit

| Susceptibility Classes | Description | Geomorphic Unit |
|---|---|---|
| High | Surrounding areas and connected to the active channel with a difference of altitude, less than one meter | Inactive channel pre dam |
| Moderatly High | Surrounding areas and/or connected with the active channel with a difference of altitude between 1 and 2 m in relation to it. | Historical floodplain |
| Moderate | Areas elevated in relation to the active channel (2-3 m). | Second terrace level (T2) |
| Moderatly Low | Areas elevated in relation to the active channel (3-4 m). | First terrace level (T1) |
| Low | Areas with a difference of altitude of more than 4 m compared to the active channel. | Fluvial aeolian plain |





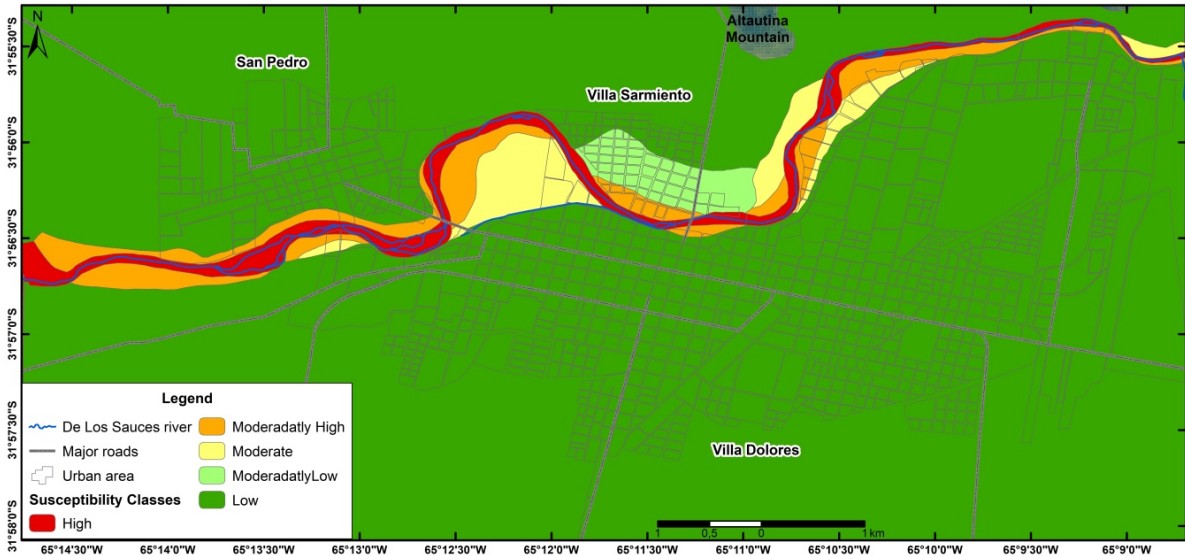

**Figure 7.** Flood susceptibility map associated with De Los Sauces River.

### -Threat Analysis

Three threat scenarios were defined according to the hydrological analysis, including flows of different magnitude and recurrence.

**Scenario 1:** Discharge values between 30 and 80 $m^3$ $s^{-1}$ are considered, which include floods of low magnitude and recurrence periods less than 10 years. These are related to the streams not intervened and to the opening of the dam sluice gates.

The threat was divided into low and very low classes according to the characteristics of the channel and intervention degree and type, which condition the flow behavior (distribution, water stage). The first class corresponds to reach 1 (R1- Table 1) which is narrower, straight, on bedrock and with the highest slope. There the flow is conducted at high velocity and show the highest stages. Towards the end of this segment, with alluvial bed and vegetation, the roughness increases and the velocity decreases, increasing the water stage. On the other hand, the very low threat was defined for the alluvial channel reach, which is wider, sinuous, multichannel and highly impacted by sediment mining (Reaches 2 and 3 – Table 1). In this case, for the estimated flows, the water stage and flow velocity are lower.

In March 2015 a scenario of these characteristics occurred. The dam was at the limit of its storage capacity, so 4 sluice gates were opened evacuating a flow close to 30 $m^3$ $s^{-1}$.

**Scenario 2:** Discharge values considered are between 100 and 300 $m^3$ $s^{-1}$. In this case, moderate magnitude flood events are included, with a recurrence of 20-30 years associated to the tributaries that drain the scarp of the Grandes Mountains and come together downstream of Boca del Río dam. The events recorded in 1981 and most recently on February 4, 2014 represent this situation. In that event, Las Tapias and Chuchiras streams evacuated an estimated discharge of 129 and 200 $m^3$




s$^{-1}$, respectively, while for De Los Sauces river a value of 130 m$^3$ s$^{-1}$ was estimated. This scenario also considers discharges
associated with the partial opening of sluice gates dam.
**Scenario 3:** Discharges of great magnitude and with recurrences greater than 50 years were estimated. This scenario would
be associated to an extraordinary event added to an inadequate management of the dam. The reservoir would reach its
maximum storage capacity evacuating a flow of approx. 1,200 m$^3$ s$^{-1}$ through the total opening of the 8 sluice gates.

239       **- Hazard Analysis**

In the Table 4 and Figs. 8, 9 and 10 the hazard maps for the three threat scenarios are showed.

241       **Table 4.** Flood hazard classes considering three threat scenarios

| GEOMORPHIC UNIT | SUSCEPTIBILITY CLASSES | THREAT (Scenario 1) | HAZARD (Scenario 1) | THREAT (Scenario 2) | HAZARD (Scenario 2) | THREAT (Scenario 3) | HAZARD (Scenario 3) |
|---|---|---|---|---|---|---|---|
| Channel | High | Moderatly Low | Moderate | Moderate | Moderatly High | High | Very High |
|  |  | Low | Moderatly Low |  |  |  |  |
| Floodplain | Moderatly High | Low | Moderatly Low | Moderate | Moderate | High | High |
| Terrace 1 (T1) | Moderate | - | - | - | - | High | Moderatly High |
| Terrace 2 (T2) | Moderatly Low | - | - | - | - | High | Moderate |
| Fluvio-aeolian Plain | Low | - | - | - | - | High | Moderatly Low |

As it is observed only for the *lower discharges,* geomorphological differences and human interventions in the pre-dam
channel, had incidence in the distribution of the threat and, therefore of the hazard, being between *moderate and moderately*
*low*, in the straight and sinuous reaches, respectively. On the other hand, flows of this magnitude are conducted in the
channel, without affecting the floodplain (Fig. 8).

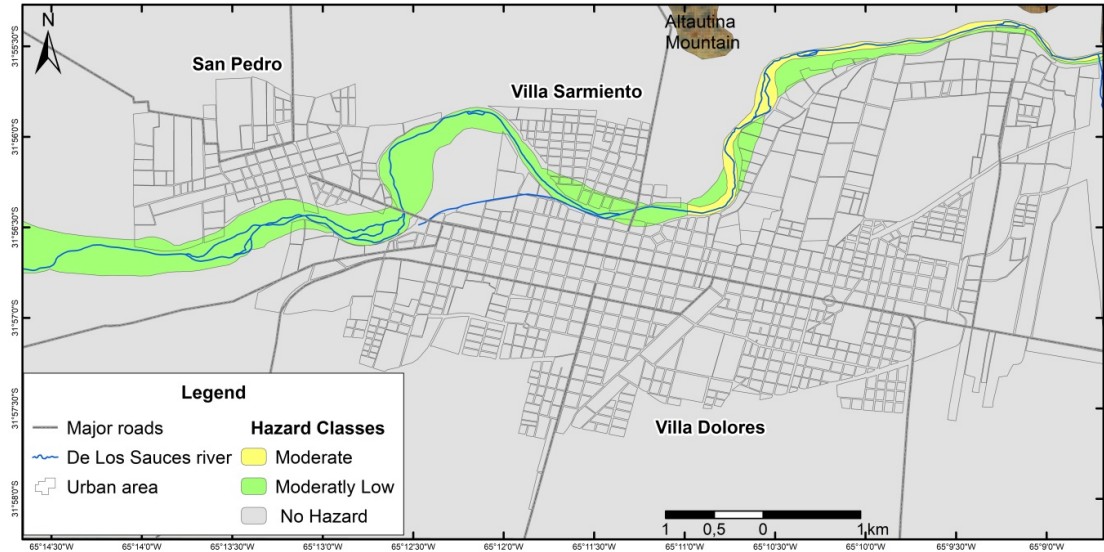

247       **Figure 8.** Flood hazard map associated with the De Los Sauces River for the first threat scenario.



For intermediate discharges (Scenario 2 –Fig. 9), *a moderately high hazard* in the pre-dam channel and *moderate hazard* for
the floodplain were estimated. It was assumed for these flow values that differences between reaches are not relevant. On the
other hand, considering the geomorphological aspects together with the occurrence of events of this magnitude, it is expected
that the evaluated flows will not affect the terraces levels due to the degree of incision of the De Los Sauces River.

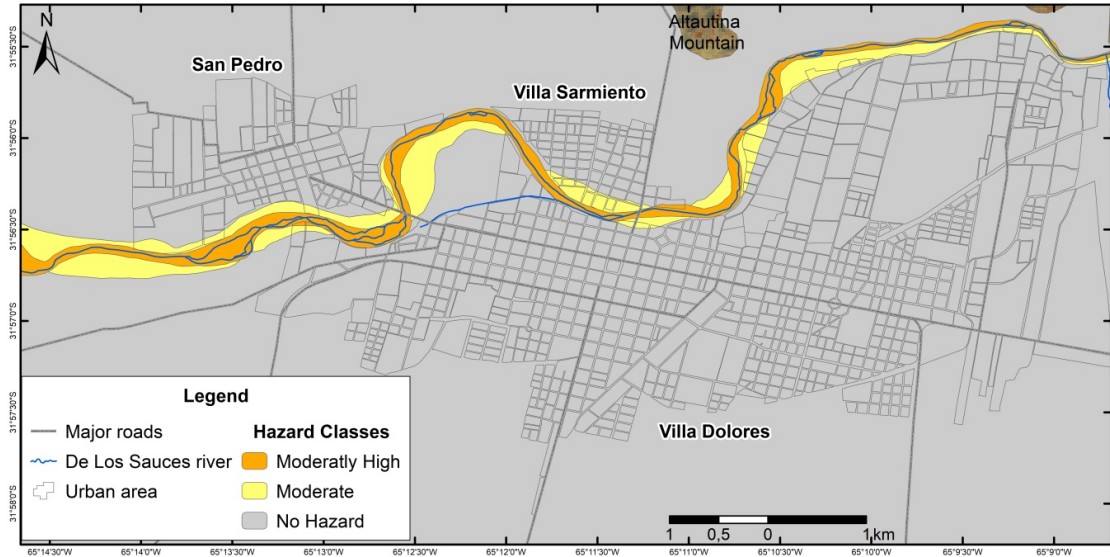

**Figure 9.** Flood hazard map associated with De Los Sauces River for the second threat scenario.

Finally, the scenario 3 involves all geomorphological environments, resulting in *very high and high hazard* in the pre-dam
and floodplain, respectively, until *moderately low* in the fluvio-aeolian plain, assuming the possible occurrence of overflows
associated with paleochannels (Fig. 10).

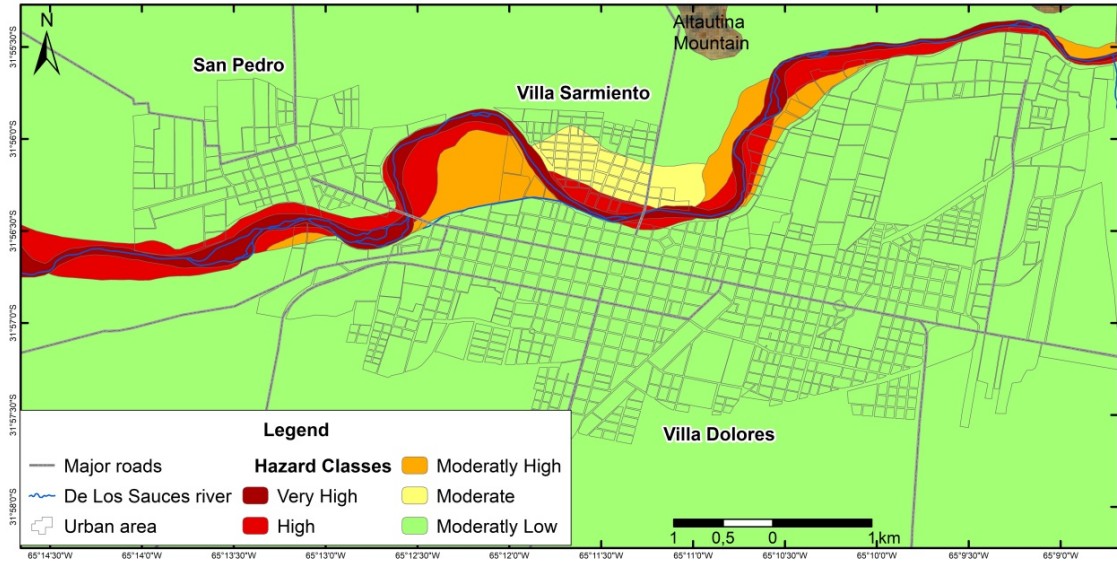

**Figure 10.** Flood hazard map associated with De Los Sauces River for the third threat scenario.



**Subsect 5.4.2. Vulnerability Analysis**
Table 5 and Fig. 11 show the defined vulnerability classes and their spatial distribution, respectively. The urban areas of the
three localities present a *high to moderate vulnerability* in the blocks with dwellings. On the other hand, rural areas with fruit
and vegetable production have *low vulnerability*.
**Table 5.** Fluvial flood vulnerability classes

| Urban Area | Vulnerability Classes |
|---|---|
| High density housing | High |
| Moderate to Low density housing | Moderate |
| Block without constructions, with services | Moderatly Low |
| Routes and Airport | High |
| **Rural Area** | Low |


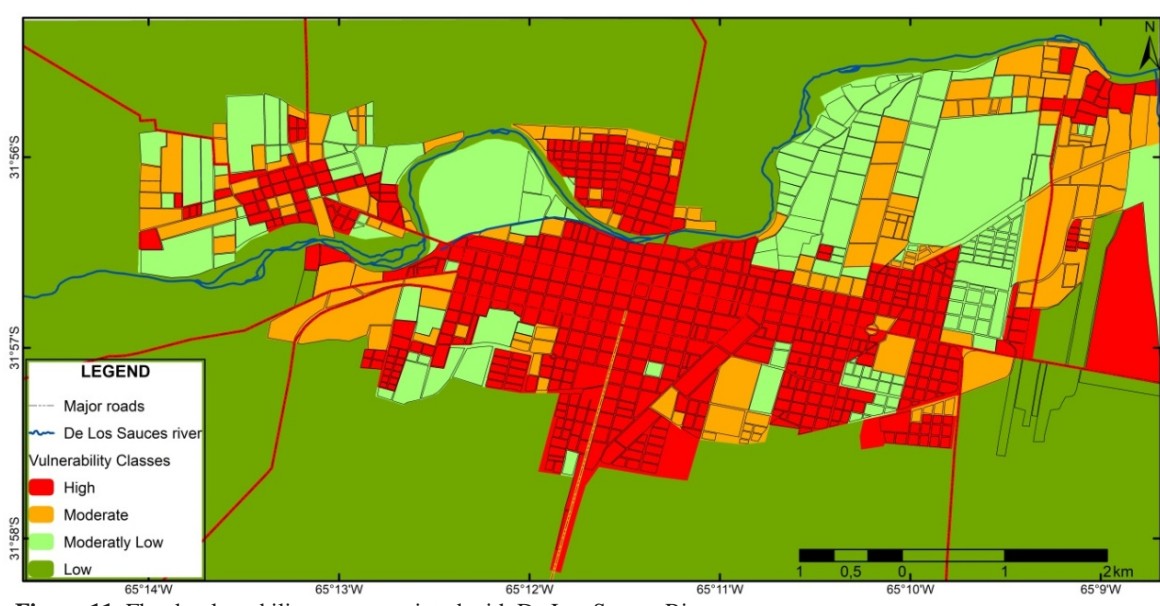

**Figure 11.** Flood vulnerability map associated with De Los Sauces River.

**Subsect 5.4.2. Risk Analysis**
Table 6 and Fig. 12, 13 and 14 show the flood risk classes associated with De Los Sauces River and the resulting maps for
the three threat scenarios considered, respectively.
**Table 6.** Flood risk clasess

| VULNERABILITY CLASSES | HAZARD CLASSES (Scenario 1) | | HAZARD CLASSES (Scenario 2) | | HAZARD CLASSES (Scenario 3) | | | | | |
|---|---|---|---|---|---|---|---|---|---|---|
| | M | ML | MH | M | VH | H | MH | M | ML | |
| **H** | MH | M | H | MH | VH | H | H | MH | M | RISK CLASSES |
| **M** | M | M | MH | M | H | MH | MH | M | M | |
| **ML** | M | ML | M | M | MH | MH | M | M | ML | |
| **L** | ML | ML | M | ML | MH | M | ML | ML | L | |

**VH:** Very High, **H:** High, **MH:** Moderatly High, **M:** Moderate, **ML:** Moderatly Low, **L:** Low.

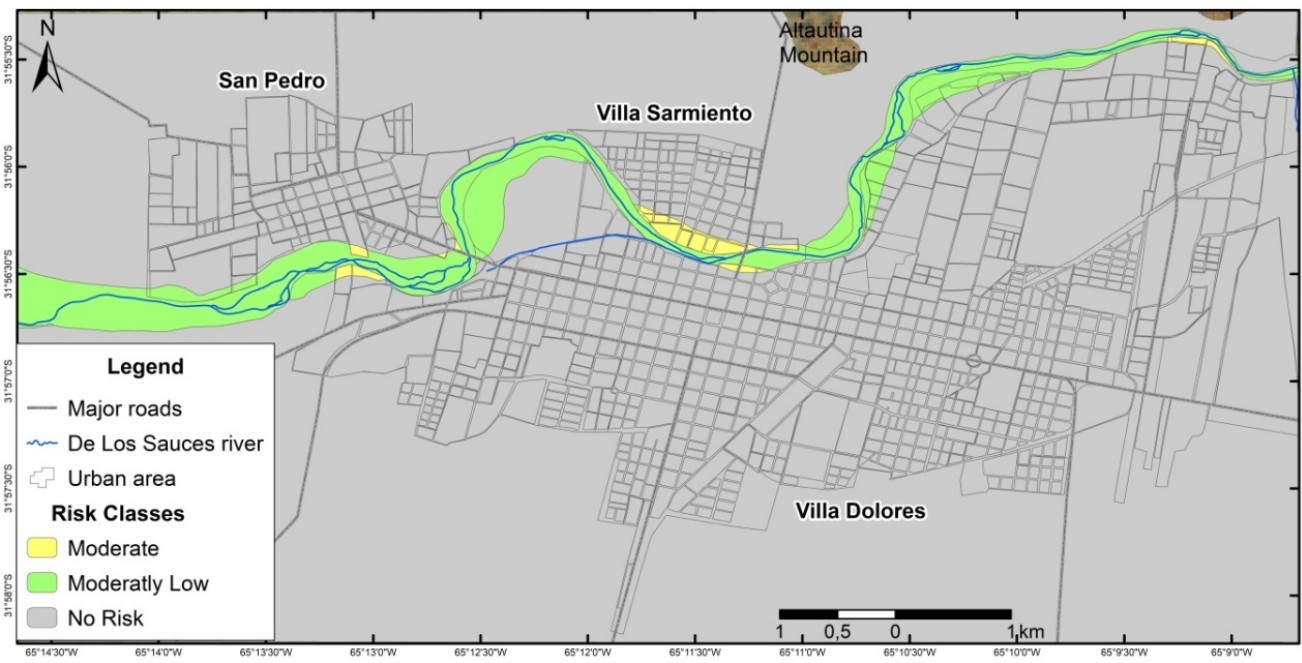

**Figure 12.** Flood risk map associated with De Los Sauces River for the first threat scenario.

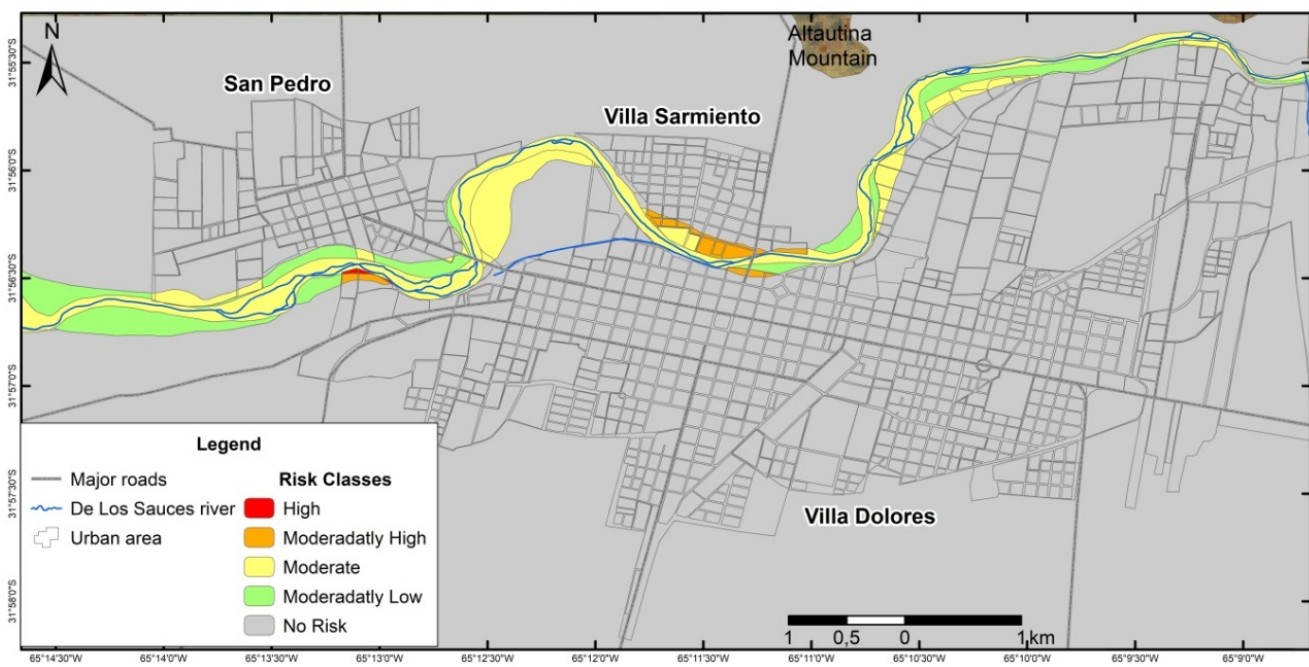

**Figure 13.** Flood risk map associated with De Los Sauces River for the second threat scenario.


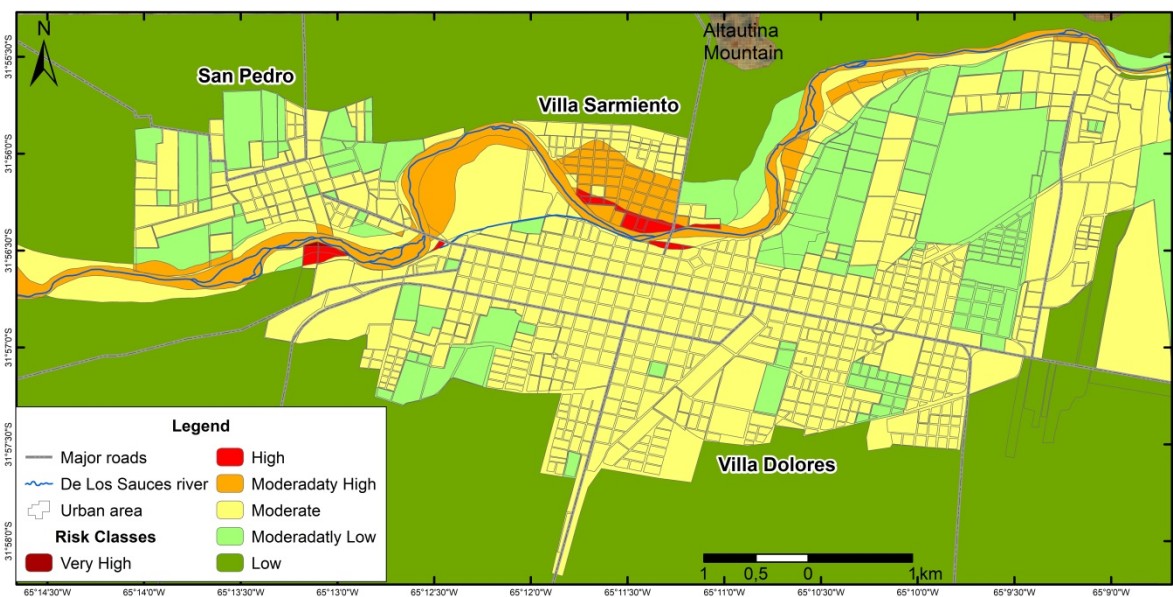

**Figure 14.** Flood risk map associated with De Los Sauces River for the third threat scenario.

In general, the areas of greatest risk are reduced (10 %) and limited to the fluvial belt, in sectors where the population forms
urban centres in the historical floodplain and in the low terraces of De Los Sauces river (Fig. 15 and 16). The fluvio aeolian
plain presents no risk or it is low in extraordinary flood events. When scenarios of increasing threat are analyzed, the areas
with risk increase from 5 % to 10 %, with risk classes from moderately low to very high and high. In turn, it is worth
emphasizing that due to the advance of urbanization on the fluvial plain there are sectors with risk, even for small floods.

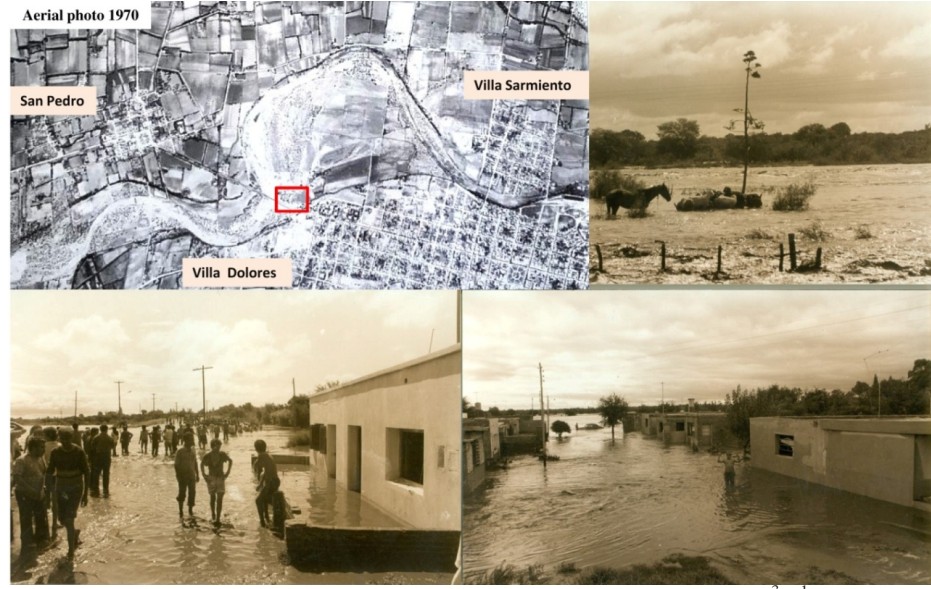

**Figure 15**. Photos of flooded sectors in 1981 with a discharge greater than 120 m$^3$ s$^{-1}$ (scenario 2).
The neighborhood seen in the photo was relocated after the flood.



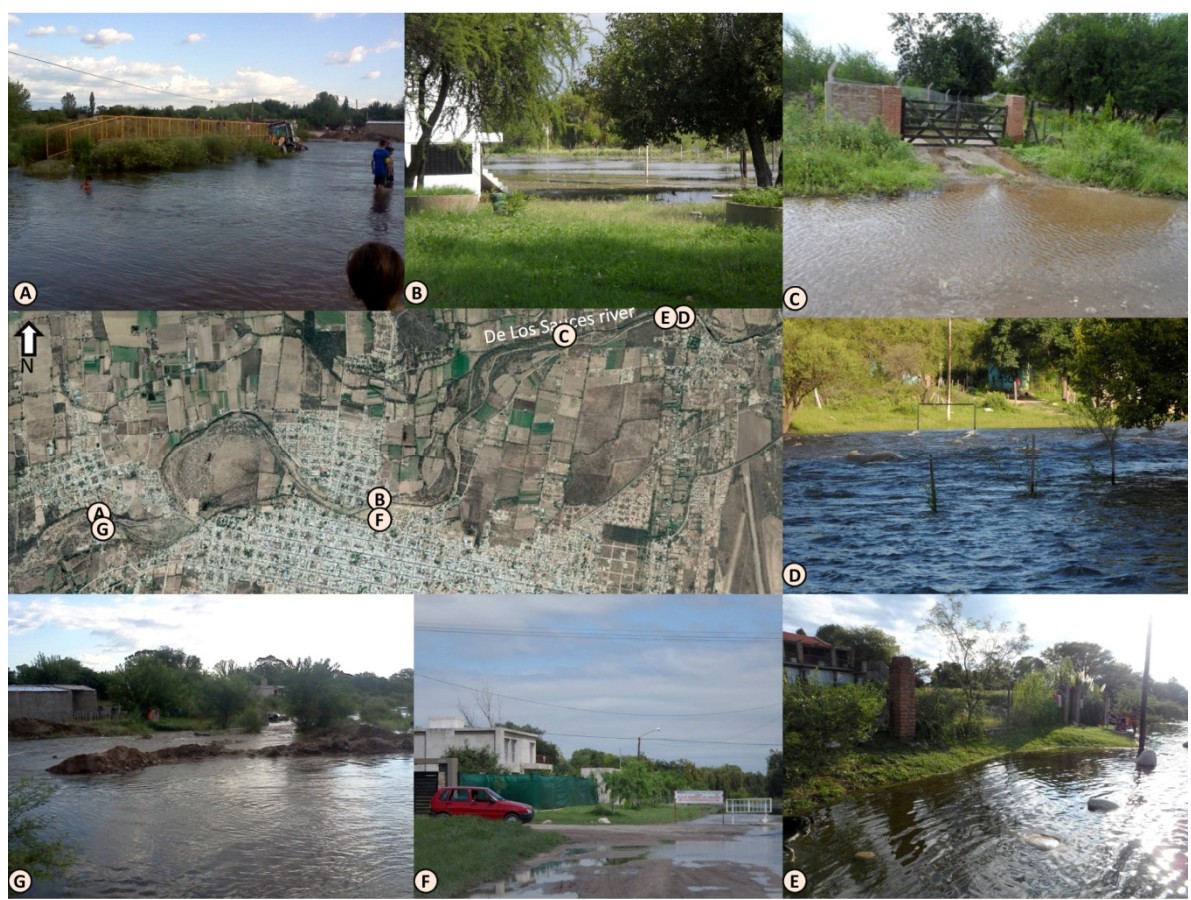

**Figure 16.** Views of zone with moderate and high flood risk in overflows occurred on February 2014 (scenario 2).

**Sect. 5. Conclusions**
The important incision degree of De Los Sauces River in its inactive alluvial fan explains the low percentage of area that has
a moderately high to high flood risk. Only due to anthropogenic causes (channels, diversions to ditches, among others) could
be considered cases where, in the event of extraordinary floods, overflows occur upstream of the urbanized areas and a risk
scenario can be presented.
The morphological and morphodynamic changes of De Los Sauces River lower-middle reach, after the Medina Allende dam
operation in 1942, caused an increase in the risk of flooding in Villa Dolores city and peripheral localities, for low to
moderate magnitude floods. Although the dam controls the greater floods of De Los Sauces River, for minor events
associated with not intervened tributaries and/or sluice gates opening, the advance of the urbanization in the pre-dam channel
and floodplain causes changes in the hazard and vulnerability. In the first case, the spatial distribution of the threat is
modified by obstructions, changes in the channel cross section, while in the second case, increases the exposure of
population and associated infrastructure.




Geomorphological studies were very effective for risk estimation, being irreplaceable for the hazard analysis, since the
geomorphological cartography is the basis for the flood susceptibility map, and the field feature recognition associated with
flood events, allow to estimate the threat magnitude and verify real hazard scenarios.
Hazard maps associated with hydrological events of low to moderate magnitude in intervened fluvial systems with dams
should constitute the base of land management plans.

**Sect. 6. Acknowledgment**: This article has been completed within the framework of research projects funded by the
Agencia Nacional de Promoción Científica y Tecnológica of Argentina (ANPCyT), the Ministerio de Ciencia y Tecnología
of the province of Córdoba (MINCyT) and the Secretaría de Ciencia y Técnica of the Universidad Nacional de Río Cuarto
(SECyT-UNRC).
The authors would like to express their gratitude to Prevenir non-governmental organization (NGO) for the provided
information and photos.
Finally, the authors thank the residents and Institutions of the towns of Villa de Las Rosas, Villa Dolores, Villa Sarmiento,
San Pedro and Las Tapias, for providing information of interest.

**Sect. 7. Reference**
Baker, V. R.: Paleoflood hydrology and extraordinary flood events, Journal of Hydrology, 15 96, 79-99, 1986.
Baker V. R.: Geomorphological understanding of floods, Geomorphology, 10, 139-156, 1994.
Baker, V. R., Kochel, R. C. and Patton, P.C. (Eds): Flood geomorphology, Wiley Interscience, New York, 503, 1988.
Barbeito, O. and Ambrosino, S.: Inundaciones repentinas en áreas serranas de Córdoba, in: inundaciones urbanas en
Argentina, Córdoba, Argentina, 205-2015, 2004.
Barbeito, O., Ambrosino, S., Bertoni, J. C. and Paoli, C. U.: Inundaciones severas por crecidas extremas, in: inundaciones
urbanas en Argentina, Córdoba, Argentina, 217-232, 2004.
Benito, G and Thorndycraft,V. R.: Palaeoflood hydrology and its role in applied hydrological sciences, Journal of
Hydrology, 313, 3–15, 2005.
Bonalumi A., Martino, R., Baldo, E., Zarco, J., Sfragulla, J., Carignano, C., Kraemer, P., Escayola, M. and Tauber, A.: Hoja
Geológica 3166-IV, Villa Dolores. Programa Nacional de Cartas Geológicas de la República Argentina, 1:250.000, Córdoba,

332  1999.

Bosisio, A.: Análisis de variables ambientales en una planicie aluvial con alta intromisión antrópica, en situación de una
crecida extraordinaria, M.S Thesis, National University of Litoral, Santa Fé, Argentina, 131 pp, 2011.
Bosque Sendra, J., Ortega Sisqués, A. and Rodríguez Espinosa, V. M.: Cartografía de riesgos naturales en América Central
con datos obtenidos desde Internet, Doc. Anàl. Geogr. 45, 41-70, 2005.
Cendrero, A.: Riesgos geológicos, ordenación del territorio y protección del medio ambiente. 1° Curso de riesgos
geológicos. Instituto Geológico Minero de España, Madrid, 327–333, 1987.



Chow, V. T.: Open-channel hydraulics. McGraw-Hill, New York, 680 pp., 1959.

Chow, V. T.: Hidrología aplicada, McGraw-Hill Interamericana, S.A., Santafé de Bogotá, 299 pp., 1994.

Dewan, A. M., Islam, M. M., Kumamoto, T. and Nishigaki, M.: Evaluating flood hazard for land-use planning in greater Dhaka of Bangladesh using remote sensing and GIS techniques, Water Resources Management, 21, 9, 1601–1612, 2007.

Domínguez Chávez, F. J., Rojas Villalobos, H. L., López González, E. and Alatorre Cejudo, L. C.: Digitalización de mapas para determinar riesgo a inundación y potencial de pérdidas económicas en el seccional de Anáhuac, Cuauhtémoc, Chihuahua: mediante Sistemas de Información Geográfica, in: Geoinformática aplicada a procesos geoambientales en el contexto local y regional: teledetección y sistemas de información geográfica, Alatorre Cejudo, L. C., Torres Olave, M. E., Rojas Villalobos, H. L., Bravo Peña, L. C., Wiebe Quintana, L. C., Gutiérrez, F. S. and López González, E. (Coord.), Dirección General de Difusión Cultural y Divulgación Científica, México,13-32, 2015.

Echevarria, K., Degiovanni, S., Blarasin, M. and Andreazzini, J.: Flash flood hazard assessment in an ungauged piedmont basin in the Sierras Pampeanas western region, province of Córdoba, Argentina, in: Advances in Geomorphology and Quaternary Studies in Argentina. Selected papers from the 6th. Congress of the Argentine Association of Geomorphology and Quaternary Studies, Serie Earth Systems Sciences, Rabassa, J. (ed.), Springer International Publishing, Cham, 181-201, 2017.

EPEC (Empresa Provincial de Energía de Córdoba): Conectados, Revista de la empresa provincial de energía en Córdoba, Año II, 20, 1-12, available at: "https://www.epec.com.ar/docs/revista/conectados_20.pdf", 2009.

Fedeski, M. and Gwilliam, J.: Urban sustainability in the presence of flood and geological hazards: The development of a GIS-based vulnerability and risk assessment methodology, Landscape and Urban Planning, 83, 50–61, 2007.

Field, C. B., Barros, V., Stocker, T.F., Qin, D., Dokken, D.J., Ebi, K.L., Mastrandrea, M.D., Mach, K.J., Plattner, G.-K., Allen, S.K., Tignor, M. and Midgley, P. M.: Managing the Risks of Extreme Events and Disasters to Advance Climate Change Adaptation, Special Report of the Intergovernmental Panel on Climate Change, Cambridge University Press, Cambridge, UK, and New York, USA, 582 pp., 2012.

Garzón Heydt, G: Las avenidas como proceso geológico, in: Geología y prevención de daños por inundaciones, Ayala Carcedo F.J. (Ed), Instituto Geológico Minero de España. Madrid, 5-55, 1985.

Gorgas, J. A., Tassile, J., Jarsún, B., Zamora, E., Bosnero, E., Lovera, E., Ravelo, A., Carnero, M., Bustos, V., Pappalardo, J., Petropulo, G., Rossetti, E. and Ledesma, M.: Los recursos naturales de la Provincia de Córdoba: Los suelos. Agencia Córdoba D.A.C. y T.S.E.M., Dirección de Ambiente-INTA Manfredi, 596 pp, 2003.

Graf, W: Downstream hydrologic and geomorphic effects of large dams on American rivers, Geomorphology, 79, 336–360, 2006.

Grant, G.: The Geomorphic Response of Gravel-bed Rivers to Dams: Perspectives and Prospects, in: Gravel-bed Rivers: Processes, Tools, Environments, First Edition, Church, M., Biron, P. M. and Roy, A. G. (eds), John Wiley & Sons Ltd., Chichester, UK, 165-181, 2012.

Gregory, K. J.: The human role in changing river channels, Geomorphology, 79, 172–191, 2006.



Hermelin, M.: Introducción a la Geología Ambiental. Geología Ambiental y Geomorfología aplicada en Colombia.
Asociación de Geocientíficos para el Desarrollo Internacional (AGID), Colombia, Reporte 16, 3-20, 1991.
Jha, A. K., Bloch, R. and Lamond, J.: Cities and flooding. A guide to integrated urban flood risk management for the 21st
century, International Bank for Reconstruction and Development and International Development Association, The World
Bank, Washington DC, 631 pp, 2012.
Koks, E. E., Jongman, B., Husby, T.G. and Botzen, W. J. W.: Combining hazard, exposure and social vulnerability to
provide lessons for flood risk management, Environmental Science & Policy, 47, 42-52, 2015.
Kron, W.: Flood risk = hazard · values · vulnerability, Water International, 30, 1, 58-68, 2005.
Ma, Y., Huang, H., Nanson, G.C., Li, Y. and Yao, W.: Channel adjustments in response to the operation of large dams: the
upper reach of the lower Yellow River, Geomorphology, 147-148, 35-48, 2012.
Masood, M. and Takeuchi, K.: Assessment of flood hazard, vulnerability and risk of mid-eastern Dhaka using DEM and 1D
hydrodynamic model, Nat Hazards, 61:757–770, 2012.
Merz, B., Thieken, A.H. And Gocht, M.: Flood risk mapping at the local scale: concepts and challenges, in: Flood Risk
Management in Europe. Innovation in Policy and Practice, Begum S., Stive, M. J.F,. Hill, J. W. (Eds), Springer, Netherlands,
387 231-251, 2007.

NU. CEPAL. CELADE: Urbanización y evolución de la población urbana de América Latina, 1950-1990, Boletín
Demográfico (Santiago), Publicaciones de las Naciones Unidas, Nº de venta S.01.II.G.91(2001), año 33, Nº especial
(LC/G.2140-P), 217 pp., 2001.
Panizza, M.: Geomorphological hazards and environmental impact: Assessment and mapping. In: Cendrero A., Lüttig
G., Wolff F.C. (eds) Planning the Use of the Earth's Surface. Lecture Notes in Earth Sciences, Springer, Berlin,
Heidelberg, 42, 101-123, 1992.
Ramos, V. A.: Las provincias geológicas del territorio argentino, in Caminos, R. (ed.) Geología Argentina, Instituto de
Geología y Recursos Minerales, Buenos Aires, Anales 29 (3), 41-96, 1999.
Sayed, M.B. and Haruyama, S.: Evaluation of flooding risk in Greater Dhaka District using satellite data and
geomorphological land classification map, Journal of Geoscience and Environment Protection, 4, 110-127, 2016.
Schmidt, J. C., and Wilcock, P. R.: Metrics for assessing the downstream effects of dams. Water Resources Research, 44,
W04404, doi:10.129/2006WR005092, 2008.
Schumm, S. A.: The Fluvial System, Wiley, New York-London, 338 pp., 1977.
Schumm, S. A.: River variability and Complexity, Cambridge University Press, New York, 220 pp., 2005.
Thornthwaite, C.: An approach towards a rational classification of climate, Geographic Review, 38, 1, 221-229, 1948.
Tucci Morelli, C.E.: Gestión de Inundaciones Urbanas, Porto Alegre, Brasil, 315 pp., 2007.
Vericat, D. and Batalla, R. J.: Efectos de las presas en la dinámica fluvial del curso bajo del río Ebro. Revista C. & G., 18, 1-
405 2, 37-50, 2004.



Vidal, C. and Romero, H.: Efectos ambientales de la urbanización de las cuencas de los ríos Bíobío y Andalién sobre los
riesgos de inundación y anegamiento de la ciudad de Concepción, in: Concepción metropolitano (AMC). Planes, procesos y
proyectos, serie GEOlibros, Pérez, L. and Hidalgo, R. (eds.), Instituto de Geografía, Pontificia Universidad Católica de
Chile, 123-149, 2010.
Vilches, O. R. and Reyes, C. M.: Riesgos naturales: evolución y modelos conceptuales, Revista Universitaria de Geografía,

411  20, 83-116, 2011

Xia, J., Deng, S., Zhou, M., Lu, J. and Xu, Q.: Geomorphic response of the Jingjiang Reach to the Three Gorges Project
operation, Earth Surface Processes and Landforms, DOI: 10.1002/esp.4043, 2016.