# Peer review of "FLOOD RISK RELATED TO A FLUVIAL SYSTEM MODIFIED BY DAMS WITH EMPHASIS ON MORPHODYNAMIC AND HYDROLOGICAL ASPECTS"

_Natural Hazards and Earth System Sciences, 2018_

## Referee Comment (RC1) · Anonymous Referee #1 · 30 Jul 2018

The paper presents an expert-driven method for a qualitative risk assessment in an urban area located after a dam in Cordoba (Argentina). In the proposed approach, the flood hazard and the potential vulnerability degree maps are combined by means of a matrix to obtain a qualitative flood risk assessment map. The main aim of the study is to evaluate the fluvial morphodynamic and hydrological effects of the dam construction on the proposed case study in terms of potential residual risk due to the different dam management scenarios (gates opening). The topic is interesting and of practical interest for flood risk management and fits in the Journal scope. The simple and parsimonious approach proposed could be a relevant contribution for the territory studied. Moreover, the overall presentation of the most part of the paper is well structured

and easy to understand by a wide audience. The figures are understandable and of a good quality. However, the paper has a number of criticalities that have not been fully addressed as exposed in the following.

General Comments Unfortunately, the novelty of the paper is not clearly explained; a state of the art is missed. Considering the several geomorphic or dem-based methods for estimation of flood hazard as well as a quite large range of qualitative and quantitative flood risk assessment tools and methods, characterized by different complexity, for different kind of floods (also considering anthropic causes) available in technical and scientific literature - some of them take into account the dynamic characteristics of risk – the authors should highlight the main new relevant scientific and/or technical questions that the paper would like to address. The main advantages and limitations of proposed method respect to technical and scientific literature approaches (so much respect to the one that considering sedimentological, morphological, and paleohydrological records) should be demonstrated and discussed in details. In particular, the transferability of the proposed methodology in other geographic region should be discussed in order to demonstrated the relevance of the contribution for the scientific community. The results presented for the proposed case study show the potential risk in case of a weak dam management or mal function. However, the conclusions don't present a guide line or a further analysis to demonstrate the importance of the proposed method in supporting risk management, i.e. in identifying the strategies (or choose among different alternatives) to reduce the effects/consequences of the dam constructions on the downstream area and limit the residual risk. Can the authors, for example, give advices to manage the dam operations or suggest structural and non-structural management actions for the area? These limitations of the paper are not really well justified and, surely, future researches need to be done on these important aspects.

Specific Comments 1. The contribution is relevant in particular for the specific case study and, thus, it should be highlighted in the title. 2. The abstract should provide

more details on novelty of the proposed research and a concise and complete summary of the approach utilized and its results. 3. A detailed state of the art of literature in the field is missing in order to show that proposed methods are valid and innovative. A comparison with other studies should be of interest for readers. 4. Several assumptions in the methodology section aren't outlined clearly; for example, I suggest to justify the choice of the number and magnitude of hazard (threat) scenarios. Moreover, the authors should justify on which basis the degree of hazard and vulnerability assessment are evaluated. 5. The performance of the risk matrix is not usually rigorous validated and, therefore, can assign identical rating to quantitatively very different risk due to its poor resolution. Finally, its subjective and not careful explained interpretation of rating can also produce errors in comparative ranking of risk areas. Can this compromise the validity and transferability of the present study in other case studies? 6. I suggest to demonstrate the usefulness of the proposed approach for risk management in the results and conclusion section. For example, it could be of interest to do a guideline for taken into account the risk dynamic considering natural and human changes or some advices to manage the dam operations or suggest structural and non-structural management actions for the area?

Other Comments 1. More information is needed for data utilized: e.g., which spatial resolution has the risk maps and if they are adequate for the scope of analysis? The morphometric conditions and their changes are quite similar in all the area? Can be the roughness values utilized listed in the text? How detailed is the topographic map utilized for the hazard assessment? 2. How the levees, bridges and all hydraulic infrastructures are taken into account? 3. How the mining and disposal of waste are taken into account in the results section for the vulnerability assessment? 4. The table 6 should be more readable if a colour ramp will be assigned to cells that represent the level of risk? 5. In the vulnerability assessment, the authors haven't considered population at risk or have estimated the potential loss of life and other environmental, economic and cultural aspects of risk. Can the proposed methodology take into account these important aspects of the flood risk assessment?

---

## Referee Comment (RC2) · Anonymous Referee #2 · 25 Sep 2018

I started by reading the manuscript to forge my own unbiased opinion. I then read the comments by reviewer 1. I agree with almost all the remarks and the criticisms by reviewer 1. I only report here below some additional comments.

For what I understand, there is a problem with the aim of the study (as it is suggested by the title of the paper) and the real focus of the study. According to the title, I was expecting a comprehensive assessment of the modifications caused by the dam construction, including a *quantitative* description of the changes in the downstream hydrological regime (e.g., Cheng et al., 2018), of the changes of flood risk (e.g., Viero et al., 2018), and of the morphodynamic processes that can be ascribed to the dam (e.g.,

Ronco et al., 2010). In the manuscript, I did not found any quantitative description of the changes in hydrological regime caused by the dam, nor a clear evaluation of the changes in flood risk between pre- and post-dam scenarios. That the fluvial system was modified by dams is quite irrelevant for the study, as only the current flood risk is actually evaluated. Indeed, the study is essentially an expert-driven, merely qualitatively risk assessment in the 10-km$^2$ areas adjacent the De Los Sauces River in Villa Dolores (Cordoba, Argentina). It is a simple case study. I do not see, from the paper, which are the repercussions of this study for the broad audience of NHESS, nor the novel methodological aspects from a scientific point of view.

In their intentions, the Authors followed the suggestion by Baker (1994), who stated that "Geomorphological flood studies, including recent advances in paleoflood hydrology, are needed as a complement to conventional hydrological approaches". Unfortunately, the conventional *quantitative* hydrological approach is substantially missing in this study. I do not see an adequate assessment of hazard, in particular from the hydrological/hydraulic point of view. For example, susceptibility is only determined based on the terrain elevation referred to the active channel (by the way, what part of the active channel is taken as a datum? The channel bottom? The banks of the main channel?). Such a procedure is in general not valid, as it does not consider the geometrical variability in a real stream. Because of different cross-sections, the presence of hydraulic works, etc., the water surface can experience substantially different rising rates at different locations for the same increase of the discharge. This means that terrain elevation alone (e.g., without the aid of hydraulic models) is not a sufficient information to determine the flood hazard of areas adjacent to the watercourse. Furthermore, the geomorphological analysis is not functional in order to determine the residual risk in the post-dam scenario, which mainly comes from sub-basins located downstream of the dam.

The points above entail that the main conclusions of the study are not supported by the analysis carried out by the Authors. Indeed, the increase in flood risk in Villa Dolores city and peripheral localities, for low to moderate magnitude floods, is not due to the construction of the dam; rather, the urbanization of floodplain areas increased the exposure/vulnerability that, in turn, led to increased flood risk. The dam, by significantly reducing flood discharges, might have induced (incautious) people to urbanize exposed areas, in a process that is similar to the so-called "levee-effect" (see, for example, Hutton et al., 2017, and references therein).

In addition, consider that geomorphology is a discipline that studies the processes that shape the land. In many parts of the paper, terms as "geomorphology" and "morphodynamic" are improperly used to indicate nothing more than the topography of the study area. For example, susceptibility is only determined based on the terrain elevation referred to the active channel. The identification of geomorphological units is merely descriptive for this study, and does not add substantial information for flood risk evaluation.

Finally, consider that the quality of the English is unacceptably low for NHESS. Please consider revising especially the grammar and the syntax, possibly with the support of a native speaker. Furthermore, the Authors should be more rigorous and careful in using technical terms.

**Specific points**

-The information content of the Abstract is very poor.
-Please consider to limit the references to grey literature (conferences, thesis), especially in Spanish. This particularly applies to the general part of the Introduction.
-l.11: Please revise the first sentence
-l.70: what is "energy" in this context?
-l.116-121: information here is insufficient. Please demonstrate that the hypothesis of uniform flow is verified at locations used to estimate the discharge. In general, discharge can be estimated with greater accuracy in control sections (if any), i.e., when subcritical to supercritical transition occurs (for example at bottom sills, narrowing cross-sections, etc.).

-l.143: a slope cannot be given in meters.

-l.215-ff.: I do not see any hydrological analysis that supports the attribution of return periods to discharge. (By the way, which scenario is to be chosen if the discharge is between 80 and 100 $m^3$/s?)

-l.239-ff.: how is the hazard obtained from susceptibility and threat?

**References**

Cheng J., Xu L., Wang X., Jiang J., You H.: Assessment of hydrologic alteration induced by the Three Gorges Dam in Dongting Lake, China. River Res Applic., 1–11, 2018. https://doi.org/10.1002/rra.3297.

Hutton N.S., Tobin G.A., Montz B.E.: The levee effect revisited: Processes and policies enabling development in Yuba County, California. J Flood Risk Management, 2018. https://doi.org/10.1111/jfr3.12469

Ronco, P., Fasolato, G., Nones, M., Di Silvio, G.: Morphological effects of damming on lower Zambezi River, Geomorphology, 115(1–2), 43-55, 2010. https://doi.org/10.1016/j.geomorph.2009.09.029.

Viero, D.P., Roder, G., Matticchio, B., Defina, A., Tarolli, P.: Floods, landscape modifications and population dynamics in anthropogenic coastal lowlands: the Polesine (northern Italy) case study, Science of The Total Environment, 2018, https://doi.org/10.1016/j.scitotenv.2018.09.121.

---

## Author Comment (AC1) · 11 Mar 2019

The authors appreciate very much the reviewer comments because they helped to improve the paper, especially in some relevant aspects related to the applied methodology and the maps results. A pdf. (supplement pdf) is presented with the most important changes made in the paper. The remaining modifications will be presented in the final version (if requested by the editor). Of course, it is important to mention that the main advantage of the proposed method is to evaluate, in a qualitative way, the flood risk in a region where the basins are not instrumented and the hydrological data are scarce and discontinuous. Currently, the scientific literature worldwide show numerous examples

(Dewan et al., 2007; Fernández and Lutz, 2010; Masood and Takeuchi, 2012; Quiroz Londoño et al., 2013; Sayed and Haruyama, 2016, among others) that use digital elevation models and its derived maps as main inputs for the flood risk mapping added to hydrological models and Geographic Information Systems. However, the free available DEM, often cannot be applied in detail areas (as in the case of our study area) due to their spatial resolution (30 m x 30 m in Argentina). On the other hand, hydrological models that usually are used for the threat evaluation require data series long enough to obtain reliable results. In Argentina, there are few equipped rivers and streams with long and reliable data series that allow estimating extraordinary flood flows and their return period. As a consequence, most of the time, the flow data considered come from instantaneous gauging or estimates (especially in flood events) made from a surveyed cross section (using sedimentological indicators, water-level marks, erosion features) and approaching velocity through the Manning equation. In this context, the proposed methodology allows the preparation of preliminary maps of flood risk, where the estimation of susceptibility is strongly based on topographical and geomorphological aspects whereas the magnitude and threat distribution were based on the information provided by the local inhabitants and that obtained from real flood events. Therefore, they are very useful to develop land use plans in areas with scarce data. In the corrected version, and as suggested by the referee, these aspects are highlighted in the introduction and conclusions sections.

In relation to the specific comments: - After the analysis of the reviewer suggestions, we agree in relation to the paper title. It is broad and does not highlight the particular case study so we decided to modify it. The new proposed title is "Flood risk assessment in an ungauged and damming stream, based on geomorphological aspects. Case study: De Los Sauces river, Córdoba, Argentina". On the other hand, the objective of the study was also modified: "to evaluate the flood risk of an alluvial floodplain intervened with dams applying a semiquantitative methodology that emphasizes on geomorphological aspects. -The introduction was modified and a detailed state of the art of the scientific literature was added, in order to compare with other works related to flood

risk assessment in the mentioned conditions. -As was mentioned previously, the lack of long series of reliable hydrological data limited the threat evaluation to real flood scenarios, except for the third scenario in which the total opening of the dam floodgates is assumed. -The susceptibility matrix was modified and more geomorphological information and land use data were incorporated, which allowed the inactive channel to be divided into different sections within the upper class, which is reflected also in the maps. -The objective of the work is not to develop a guideline for taking into account the risk dynamic, although the suggestion is appreciated.

Other Comments: -For this paper, the mining and solid waste disposal were considered in the susceptibility evaluation and in the threat scenarios. Due to the lack of machinery and installed infrastructure, mining was not considered in the vulnerability analysis. On the other hand, the waste disposal, in this case, is not considered very relevant taking into account the involved volumes. -The colors in Tables 4 and 6 were removed to avoid the reader confusion. -The vulnerability analysis was made taking into consideration the population density and main routes access to the localities. Both were considered especially relevant in this case. Although there are numerous and detailed studies on social vulnerability in the flood risk assessment (Birkmann, 2007; Fekete, 2009; Koks et al., 2015; Ahmed et al., 2019, among others), the objective of this work is not to evaluate social, economic and cultural aspects of the vulnerability but to establish a first vulnerability approximation to estimate the risk. It is not ruled out that in future studies a more detailed vulnerability study could be carried out by social experts.

REFERENCES Ahmed T., El-Zein, A., Tonmoy, F.N., Maggi, F.and Chung, K.S.K. Flood Exposure and Social Vulnerability for Prioritizing Local Adaptation of Urban Storm Water Systems. In: Mathew J., Lim C., Ma L., Sands D., Cholette M., Borghesani P. (eds) Asset Intelligence through Integration and Interoperability and Contemporary Vibration Engineering Technologies. Lecture Notes in Mechanical Engineering. Springer, Cham, 2019. Birkmann, J.: Risk and vulnerability indicators at different scales: Applicability, usefulness and policy implications, Environmental Hazards 7, 1, 20-31, 2007. Dewan,

A. M., Islam, M. M., Kumamoto, T. and Nishigaki, M.: Evaluating flood hazard for land-use planning in greater Dhaka of Bangladesh using remote sensing and GIS techniques, Water Resources Management, 21, 9, 1601–1612, 2007. Fekete, A.: Validation of a social vulnerability index in context to river-floods in Germany, Natural Hazards and Earth System Sciences, 9, 393–403, 2009. Fernández, D.S. and Lutz M.A.: Urban flood hazard zoning in Tucumán Province, Argentina, using GIS and multicriteria decision analysis, Engineering Geology 111, 90–98, 2010. Koks E.E., Jongman, B., Husby, T.G. and Botzen W.J.W.: Combining hazard, exposure and social vulnerability to provide lessons for flood risk management, Environmental science and policy 47, 42-52, 2015. Masood, M. and Takeuchi, K.: Assessment of Flood Hazard, Vulnerability and Risk of Mid-Eastern Dhaka Using DEM and 1D Hydrodynamic Model, Natural Hazards, 61, 757-770, 2012. Quiroz Londoño, O. M., Grondona , S. I., Massone, H. E., Farenga, M., Martínez, G. and Martínez, D. E.: Modelo de anegamiento y estrategia de predicción-prevención del riesgo de inundación en áreas de llanura: el sudeste de la provincia de Buenos Aires como caso de estudio, Revista Internacional de Ciencia y Tecnología de la Información Geográfica (GeoFocus) 13, 1, 76-98, 2013. Sayed, M.B. and Haruyama, S.: Evaluation of flooding risk in Greater Dhaka District using satellite data and geomorphological land classification map, Journal of Geoscience and Environment Protection, 4, 110-127, 2016.

Please also note the supplement to this comment:
https://www.nat-hazards-earth-syst-sci-discuss.net/nhess-2018-162/nhess-2018-162-AC1-supplement.pdf

---

## Author Comment (AC2) · 11 Mar 2019

The authors appreciate very much the reviewer comments because they helped to improve the paper, especially in some relevant aspects related to the applied methodology and the maps results. A pdf. (supplementary pdf) is presented with the most important changes made in the paper. The remaining modifications will be presented in the final version (if requested by the editor). The answer to referee # 1 is attached taking into account that you agreed with almost all the observations and suggestions. On the other hand, we answer to his valuable contributions: After the analysis of the reviewer suggestions, we agree that the title, in part, does not reflect the content and

data presented in the paper. Unfortunately, the reviewer generated expectations not meets by the paper and make suggestions to the work in this regard. As mentioned previously, the hydrological data are scarce and discontinuous and the analysis of the threat was limited to real events, except for the third scenario. On the other hand, more geomorphological data were added to support the considered susceptibility values, added to the topographic data that appear in the submitted paper and that was observed by the referee.

Specific points Although the use of Spanish and gray literature (conferences, thesis) is not recommended, we believe it is important to mention it to show that the works done are very limited in the study area. Therefore, the presented work is a flood risk preliminary assessment in an area where the watersheds are not instrumented, hydrological data are scarce, there is an advance of urbanization on the floodplain and there are institutions interested in order to solve these problems. l.70. In disciplines such as geomorphology and sedimentology the term is used e.g. high and low energy processes, referring to flows of different velocity that generate deposits of different grain sizes. In this context, the term "energy" is used. l.143. The appropriate term is height difference (expressed in m) and not slope (% or degree). The error was not conceptual but translation. l.215 ff. As mentioned, the obtained eyewitness reports from local residents and water-level marks (considering vegetation, sediment distribution, erosion features) were a very important information source on the hydrological events. l. 239. As mentioned in the methodology, the Hazard represents the Susceptibility or natural fragility of a región exposed to a certain Threat. The susceptibility includes the geological, geomorphological, lithological, hydrological, geotechnical aspects, among others, that together determine the behavior of an area in front of a natural process (Panizza, 1992), whereas the Threat, according to Hermelin (1991), is the probability of occurrence of a potentially destructive phenomenon within a specific time period for a specific area. Therefore, the hazard results presented in Table 4 arise from crossing, in this case qualitatively, of the susceptibility class with the threat.

Answer to referee # 1: Of course, it is important to mention that the main advantage of the proposed method is to evaluate, in a qualitative way, the flood risk in a region where the basins are not instrumented and the hydrological data are scarce and discontinuous. Currently, the scientific literature worldwide show numerous examples (Dewan et al., 2007; Fernández and Lutz, 2010; Masood and Takeuchi, 2012; Quiroz Londoño et al., 2013; Sayed and Haruyama, 2016, among others) that use digital elevation models and its derived maps as main inputs for the flood risk mapping added to hydrological models and Geographic Information Systems. However, the free available DEM, often cannot be applied in detail areas (as in the case of our study area) due to their spatial resolution (30 m x 30 m in Argentina). On the other hand, hydrological models that usually are used for the threat evaluation require data series long enough to obtain reliable results. In Argentina, there are few equipped rivers and streams with long and reliable data series that allow estimating extraordinary flood flows and their return period. As a consequence, most of the time, the flow data considered come from instantaneous gauging or estimates (especially in flood events) made from a surveyed cross section (using sedimentological indicators, water-level marks, erosion features) and approaching velocity through the Manning equation. In this context, the proposed methodology allows the preparation of preliminary maps of flood risk, where the estimation of susceptibility is strongly based on topographical and geomorphological aspects whereas the magnitude and threat distribution were based on the information provided by the local inhabitants and that obtained from real flood events. Therefore, they are very useful to develop land use plans in areas with scarce data. In the corrected version, and as suggested by the referee, these aspects are highlighted in the introduction and conclusions sections.

In relation to the specific comments: - After the analysis of the reviewer suggestions, we agree in relation to the paper title. It is broad and does not highlight the particular case study so we decided to modify it. The new proposed title is "Flood risk assessment in an ungauged and damming stream, based on geomorphological aspects. Case study: De Los Sauces river, Córdoba, Argentina". On the other hand, the objective of the

study was also modified: "to evaluate the flood risk of an alluvial floodplain intervened with dams applying a semiquantitative methodology that emphasizes on geomorphological aspects. -The introduction was modified and a detailed state of the art of the scientific literature was added, in order to compare with other works related to flood risk assessment in the mentioned conditions. -As was mentioned previously, the lack of long series of reliable hydrological data limited the threat evaluation to real flood scenarios, except for the third scenario in which the total opening of the dam floodgates is assumed. -The susceptibility matrix was modified and more geomorphological information and land use data were incorporated, which allowed the inactive channel to be divided into different sections within the upper class, which is reflected also in the maps. -The objective of the work is not to develop a guideline for taking into account the risk dynamic, although the suggestion is appreciated.

Other Comments: -For this paper, the mining and solid waste disposal were considered in the susceptibility evaluation and in the threat scenarios. Due to the lack of machinery and installed infrastructure, mining was not considered in the vulnerability analysis. On the other hand, the waste disposal, in this case, is not considered very relevant taking into account the involved volumes. -The colors in Tables 4 and 6 were removed to avoid the reader confusion. -The vulnerability analysis was made taking into consideration the population density and main routes access to the localities. Both were considered especially relevant in this case. Although there are numerous and detailed studies on social vulnerability in the flood risk assessment (Birkmann, 2007; Fekete, 2009; Koks et al., 2015; Ahmed et al., 2019, among others), the objective of this work is not to evaluate social, economic and cultural aspects of the vulnerability but to establish a first vulnerability approximation to estimate the risk. It is not ruled out that in future studies a more detailed vulnerability study could be carried out by social experts. REFERENCES Ahmed T., El-Zein, A., Tonmoy, F.N., Maggi, F.and Chung, K.S.K. Flood Exposure and Social Vulnerability for Prioritizing Local Adaptation of Urban Storm Water Systems. In: Mathew J., Lim C., Ma L., Sands D., Cholette M., Borghesani P. (eds) Asset Intelligence through Integration and Interoperability and

[Figure]

Contemporary Vibration Engineering Technologies. Lecture Notes in Mechanical Engineering. Springer, Cham, 2019. Birkmann, J.: Risk and vulnerability indicators at different scales: Applicability, usefulness and policy implications, Environmental Hazards 7, 1, 20-31, 2007. Dewan, A. M., Islam, M. M., Kumamoto, T. and Nishigaki, M.: Evaluating flood hazard for land-use planning in greater Dhaka of Bangladesh using remote sensing and GIS techniques, Water Resources Management, 21, 9, 1601–1612, 2007. Fekete, A.: Validation of a social vulnerability index in context to river-floods in Germany, Natural Hazards and Earth System Sciences, 9, 393–403, 2009. Fernández, D.S. and Lutz M.A.: Urban flood hazard zoning in Tucumán Province, Argentina, using GIS and multicriteria decision analysis, Engineering Geology 111, 90–98, 2010. Koks E.E., Jongman, B., Husby, T.G. and Botzen W.J.W.: Combining hazard, exposure and social vulnerability to provide lessons for flood risk management, Environmental science and policy 47, 42-52, 2015. Masood, M. and Takeuchi, K.: Assessment of Flood Hazard, Vulnerability and Risk of Mid-Eastern Dhaka Using DEM and 1D Hydrodynamic Model, Natural Hazards, 61, 757-770, 2012. Quiroz Londoño, O. M., Grondona , S. I., Massone, H. E., Farenga, M., Martínez, G. and Martínez, D. E.: Modelo de anegamiento y estrategia de predicción-prevención del riesgo de inundación en áreas de llanura: el sudeste de la provincia de Buenos Aires como caso de estudio, Revista Internacional de Ciencia y Tecnología de la Información Geográfica (GeoFocus) 13, 1, 76-98, 2013. Sayed, M.B. and Haruyama, S.: Evaluation of flooding risk in Greater Dhaka District using satellite data and geomorphological land classification map, Journal of Geoscience and Environment Protection, 4, 110-127, 2016.

Please also note the supplement to this comment:
https://www.nat-hazards-earth-syst-sci-discuss.net/nhess-2018-162/nhess-2018-162-AC2-supplement.pdf
* * *
2018-162, 2018.

**Supplement:**

The main changes made in the work are presented based on the referee suggestions. The remaining modifications will be presented in the final version.

**Proposed title**: "Flood risk assessment in an ungauged and damming stream, based on geomorphological aspects. Case study: De Los Sauces river, Córdoba, Argentina".

**Proposed objective is**: "to evaluate the flood risk of an alluvial floodplain intervened with dams applying a semi quantitative methodology that emphasizes on geomorphological aspects.

**Sect. 5. Results**

**Subsect 5.1.Geomorphological and Topographic Characterization**

The study area is located in the proximal sector of the alluvial paleofan (Neogene-Quaternary) of the De Los Sauces River, where the current course presents different incision degree and varied development of the fluvial belt.

In this context, five geomorphological units were recognized (Fig.4).

[Figure]

**Figure 4.**Geomorphological Map of the study area.

**I-Fluvio-aeolian Plain:** corresponds to the oldest surface of the paleofan. The relief is very gently undulated, where the loessical layers and longitudinal dunes are interdigitated and/or overlaying the paleochannels and overflow lobes of the De Los Sauces River. It has a slope to the west on the order of (0.55)

and a height with respect to the active channel that decreases in that direction from 8 to 2-3 m approximately when entering the middle sector of the paleofan.

   **II-De Los Sauces River fluvial belt:** it extends downstream from Piedra Pintada location and is the result of different incision pulses from De Los Sauces River during the Upper Holocene to the Present. It has a width between 300 and 1,500 m, associated with straight (bedrock) and meandering/braided (alluvial) channel reaches. It includes two discontinuous levels of terraces (III.D and E-Fig. 4) and a small floodplain (III.C-Fig. 4) associated with the active channel (III.A-Fig. 4).The oldest terrace level (T1) has a height of 3-4 m and the lower level (T2) of 2-3 m, above the channel bottom. The streambed sediment are coarse gravelly (cobbles, boulders) – sandy (very coarse) and poorly sorted. The grain size decreasing slightly downstream of bedrock reach (< boulders).

The channel of De Los Sauces River shows variability, not only linked to geological controls but as a result of the operation of the Medina Allende dam and human occupation. In fact, the reduction of waterflow seems to have an immediate effect downstream by initially fostering the sediment deposition. Subsequently, the total interception of sediment by the dam slowly takes over and inverts this tendency. A slightly smaller aggradation (or slightly larger degradation) rate with respect to the natural conditions (no dams) seems to represent the dominant effect of damming in the long term evolution of De Los Sauces river channel. The deposition of fine sandy sediments in the streambed increases, and consequently, grain size sorting decrease. In general, the bedrock segment do not exhibit significant morphological changes (except the channel width), while the alluvial channel lost its braided behavior, although it maintained its sinuosity, prevailing a semiconfined single channel with and erosive behavior. The channel width was reduced up to 85%, generating a historical floodplain. The channel was segmented in four parts considering the most relevant morphological and morphometric characteristics in pre and post dam conditions (Table 1).

**Table 1.** Most relevant morphological and morphometric characteristics of the channel in pre and post dam conditions.

| U.II.3 Active Channel | | Types of river channel | Channel Patterns | Height Bank (m) | Length (km) | Slope (%) | Width (m) | Width channel reduction (%) 1970-2017 |
|---|---|---|---|---|---|---|---|---|
| **R1** | Pre-dam | Bedrock | Straight Single Channel (SI:1.1 | 3-4 | --0.5 | 0.5 | 15-20 | 50-75 |
| | Post-dam | | | | 0.5 | | 5-7 | |
| **R2** | Pre-dam | Alluvial | Straight Single Channel (SI:1.1) | 3-4 | 3.5 | 0.4 | 120-150 | 90 |
| | Post-dam | | | | | | 12-16 | |
| **R3** | Pre-dam | Alluvial | Meandering with overlay braided, mobile bars | 3-6 | 5.8 | 0.32 | 40 | 80 |

| R | Period | Substrate | Description | | | | | |
|---|--------|-----------|-------------|---|---|---|---|---|
| | Post-dam | Alluvial | Meandering (SI: 1.6) Single channel dominate and secondary channel, locally (BI: 2), very vegetated and stable bars | | | | 8 | |
| **R4** | Pre-dam | Alluvial | Meandering Braided | 2-5 | 4 | 0.32 | 70-60 | 85 |
| | Post-dam | Alluvial | Multichannel (SI: 1.2, BI: 4) Irregular, erosive and secondary channels. Ponds presence by mining | | | | 10-12 | |

SI: Sinuosity Index, BI: Braiding Index

**Subsect 5.4. Analysis of flood risk**

**Subsect 5.4.1 Fluvial flood hazard**

**-Susceptibility analysis**

Five susceptibility classes were defined (Table 3) which were evaluated in each geomorphological unit (Fig. 7). As can be observed in the map, the susceptible zones are those located in the most modern fluvial belt. Taking into account that it is incised in the paleo alluvial fan and then deepened, these zones have very low susceptibility.

**Table 3.** Susceptibility classes evaluated for each geomorphological unit

| Susceptibility Classes | Geomorphic Unit | | | Land Use |
|------------------------|-----------------|---|---|----------|
| High | Inactive channel Post dam | + ... - | Reach R1 | Engineering works (bridges, walkway, fords, duct and small dike), local roads, mining |
| | | | Reach R2 | |
| | | | Reach R3 | |
| | | | Reach R4 | |
| Moderately High | Historical floodplain | | | Mining, motocross circuit |
| Moderate | Second terrace level (T2) | | | Mining, irrigation channels |
| Moderately Low | First terrace level (T1) | | | - |
| Low | Fluvial aeolian plain | | | - |

[Figure]

**Figure 7.** Flood susceptibility map associated with De Los Sauces River.

**-Threat Analysis**

Three threat scenarios were defined according to the hydrological analysis, including flows of different magnitude and recurrence. On the other hand, for the scenarios 1 and 2, the threat was subdivided into two classes according to the channel characteristics and intervention degree and type, which condition the flow behavior (distribution, waterstage). The highest class corresponds to reaches R1-R2 (Tables 1 and 4) which are narrower, straight, on bedrock/alluvial and with the highest slope. There the flow is conducted at high velocity and show the highest stages. The lowest class was defined for the alluvial channel reach, which is wider, sinuous, multichannel and highly impacted by mining (Reaches 3 and 4 – Tables 1and 4). In this case, the roughness increases, the water stage and flow velocity are lower.

**Scenario 1:** Discharge values between 30 and 80 $m^3s^{-1}$ are considered, which include floods of low magnitude and recurrence periods less than 10 years. These are related to the streams not intervened and to the opening of the dam sluice gates.

In March 2015 a scenario of these characteristics occurred. The dam was at the limit of its storage capacity, so 4 sluice gates were opened evacuating a flow close to 30 $m^3s^{-1}$.

**Scenario 2:** Discharge values considered are between 80 and 300 $m^3s^{-1}$. In this case, moderate magnitude flood events are included, with a recurrence of 20-30 years associated to the tributaries that drain the scarp of the Grandes Mountains and come together downstream of Boca del Río dam. The events recorded in 1981 and most recently on February 4, 2014 represent this situation. In that event, Las Tapias and Chuchiras streams evacuated an estimated discharge of 129 and 200 $m^3s^{-1}$, respectively, while for De Los Sauces river a value of 130 $m^3s^{-1}$ was estimated. This scenario also considers discharges associated with the partial opening of sluice gates dam.

**Scenario 3:** Discharges of great magnitude and with recurrences greater than 50 years were estimated. This scenario would be associated to an extraordinary event added to an inadequate management of the dam. The reservoir would reach its maximum storage capacity evacuating a flow of approx. 1,200 $m^3s^{-1}$ through the total opening of the 8 sluice gates.

**- HazardAnalysis**

In the Table 4 and Figs. 8, 9 and 10 the hazard maps for the three threat scenarios are showed.

**Table 4.** Flood hazard classes considering three threat scenarios

| GEOMORPHIC UNIT | SUSCEPTIBILITY CLASSES | | THREAT (Scenario 1) | HAZARD (Scenario 1) | THREAT (Scenario 2) | HAZARD (Scenario 2) | THREAT (Scenario 3) | HAZARD (Scenario 3) |
|---|---|---|---|---|---|---|---|---|
| Channel | High | R1 | Moderately Low (R1-R2) | Moderately High (R1-R2 | Moderately High (R1-R2 | High (R1-R2) | Very High | Very High |
| | | R2 | | | | | | |
| | | R3 | Low (R3-R4) | Moderate (R3-R4) | Moderate (R3-R4) | Moderately hight (R3-R4) | | |
| | | R4 | | | | | | |
| Floodplain | Moderately High | | Very Low | Moderately Low | Moderately low | Moderate | High | High |
| Terrace 2 (T2) | Moderate | | - | - | - | - | Moderately High | Moderately High |
| Terrace 1 (T1) | Moderately Low | | - | - | - | - | Moderate | Moderate |
| Fluvio-aeolian Plain | Low | | - | - | - | - | Moderatey Low | Moderately Low |

[Figure]

**Figure 8.** Flood hazard map associated with the De Los Sauces River for the first threat scenario.

[Figure]

**Figure 9.** Flood hazard map associated with De Los Sauces River for the second threat scenario.

[Figure]

**Figure 10.** Flood hazard map associated with De Los Sauces River for the third threat scenario.

The vulnerability map is the same as that presented in the submitted paper. Therefore flood risk maps will change according to the flood hazard maps. They will be presented in the final version.